# Large-scale sensitivities of groundwater and surface water to groundwater withdrawal

*Marc F.P. Bierkens[1,2,#], Edwin H. Sutanudjaja[1] and Niko Wanders[1]*

[1]Department of Physical Geography, Utrecht University, P.O. Box 80115, 3508 TC Utrecht, The Netherlands
[2]Unit Soil and Groundwater Systems, Deltares, P.O. Box 85467, 3508 AL Utrecht, The Netherlands
# Correspondence to: m.f.p.bierkens@uu.nl

## Abstract

Increasing population, economic growth and changes in diet have dramatically increased the demand for food and water over the last decades. To meet increasing demands, irrigated agriculture has expanded into semi-arid areas with limited precipitation and surface water availability. This has greatly intensified the dependence of irrigated crops on groundwater withdrawal and caused a steady increase of groundwater withdrawal and groundwater depletion. One of the effects of groundwater pumping is the reduction in streamflow through capture of groundwater recharge, with detrimental effects on aquatic ecosystems. The degree to which groundwater withdrawal affects streamflow or groundwater storage depends on the nature of the groundwater-surface water interaction (GWSI). So far, analytical solutions that have been derived to calculate the impact of groundwater on streamflow depletion involve single wells and streams and do not allow the GWSI to shift from connected to disconnected, i.e. from a situation with two-way interaction to one with a one-way interaction between groundwater and surface water. Including this shift and also analyse the effects of many wells, requires numerical groundwater models that are expensive to setup. Here, we introduce an analytical framework based on a simple lumped conceptual model that allows to estimate to what extent groundwater withdrawal affects groundwater heads and streamflow at regional scales. It accounts for a shift in GWSI, calculates at which critical withdrawal rate such a shift is expected and when it is likely to occur after withdrawal commences. It also provides estimates of streamflow depletion and which part of the groundwater withdrawal comes out of groundwater storage and which parts from a reduction in streamflow. After a local sensitivity analysis, the framework is combined with parameters and inputs from a global hydrological model and subsequently used to provide global maps of critical withdrawal rates and timing, the areas where current withdrawal exceeds critical limits, and maps of groundwater depletion and streamflow depletion rates that result from groundwater

withdrawal. The resulting global depletion rates are compared with estimates from in situ-observations, regional and global groundwater models and satellites. Pairing of the analytical framework with more complex global hydrological models presents a screening tool for fast first-order assessments of regional-scale groundwater sustainability, and for supporting hydroeconomic models that require simple relationships between groundwater withdrawal rates and the evolution of pumping costs and environmental externalities.

## 1. Introduction

Increasing population, economic growth and changes in diet have dramatically increased the demand for food and water over the last decades (Godfray et al., 2010). To meet increasing demands, irrigated agriculture has expanded into semi-arid areas with limited precipitation and surface water (Siebert et al., 2015). This has greatly intensified the dependence of irrigated crops on groundwater withdrawal (Wada et al., 2012) and caused a steady increase of groundwater depletion rates (Wada and Bierkens, 2019). Recent estimates of current groundwater withdrawal range approximately between 600-1000 $km^3$ $yr^{-1}$ leading to estimated depletion rates of 150-400 $km^3$ $yr^{-1}$ (Wada, 2016).

Groundwater that is pumped comes either out of storage, from reduced groundwater discharge or from reduction of evaporation fed from below by groundwater through capillary rise and/or phreatophytes (Theis, 1940; Alley et al. 1999; Bredehoeft, 2002); Konikow and Leake, 2014). Thus, extensive groundwater pumping not only leads to groundwater depletion (Wada et al., 2010) but also to a reduction in streamflow (Wada et al., 2013; Mukherjee et al., 2019; De Graaf et al., 2019; Jasechko et al., 2021) and desiccation of wetlands and groundwater dependent terrestrial ecosystems (Runhaar et al 1997; Shafroth et al., 2000; Elmore et al 2006; Yin et al 2018). However, the effect of groundwater pumping on groundwater depletion and surface water depletion heavily depends on the nature of the interaction between groundwater and surface water. Limiting ourselves to phreatic groundwater systems and following Winter et al. (1998), a distinction can be made between gaining streams, loosing streams and disconnected loosing streams, depending on the position of the free groundwater surface with respect to the surface water level and the bottom of the stream (Figure 1). Since groundwater pumping affects groundwater levels, it can move a stream from gaining to losing to disconnected and loosing, which, in turn, affects the way that groundwater pumping affects streamflow.

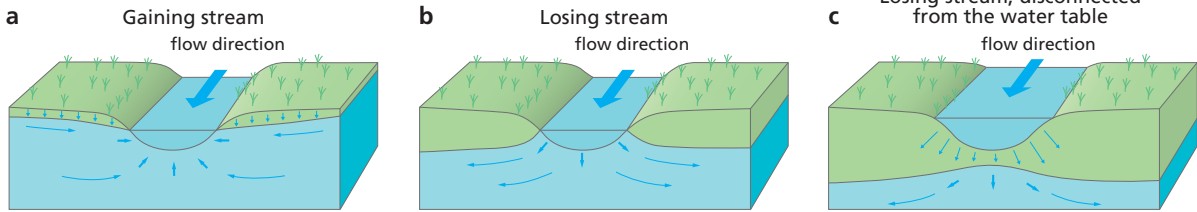

*Fig. 1. Groundwater-streamflow interaction: (a) gaining stream; (b) losing stream; (c)losing stream disconnected from the water table; modified from Winter at al. (1998); credit to the United States Geological Survey.*

Based on the above, Bierkens and Wada (2019) define two stages of groundwater withdrawal in phreatic aquifers. In stage 1, groundwater withdrawal is such that the water table remains connected with the surface water system (Figure 1a, b). Upon pumping, groundwater initially comes out of storage and groundwater levels decline. However, as groundwater levels decline around a well, the well attracts more of the recharge that would otherwise end up in the stream until a new equilibrium is reached where all of the pumped water comes out of captured streamflow. In a stage 1 withdrawal regime, withdrawal can be considered as physically stable, where groundwater depletion is limited and groundwater withdrawal mostly diminishes streamflow and evaporation. Depending on the groundwater level, one could further distinguish between gaining (Figure 1a) and loosing (Figure 1b) streams. This is important when considering the quality of pumped groundwater as in case of a losing stream surface water ends up in the well. In a stage 2 withdrawal regime, groundwater withdrawal is so large that groundwater levels fall below the bottom of the stream (Figure 1c). In that case, a further decline of the groundwater level hardly increases infiltration from the stream to the aquifer. Thus, in stage 2, groundwater withdrawal in excess of recharge and (constant) stream water infiltration is physically unstable and as a result leads to groundwater depletion and does not impact streamflow further if pumping rates increase.

From the above it follows that there is a critical transition between stage 1 and stage 2 groundwater withdrawal that depends on groundwater withdrawal rate. In reality, this transition is less abrupt. Right after the water table is just below the river bottom, negative pressure heads occur below the river bed while the soil is fully or partly saturated. Wang et al. (2015) show experimentally and theoretically that a full disconnection, i.e. the water table has no impact on the infiltration flux, occurs only when the depth of the groundwater table below the stream becomes larger than the stream water depth. Another reason that this transition does not occur abruptly is that multiple surface water bodies in the surroundings of

groundwater wells differ in depth depending on stream order and location in the river basin.
We also note that that in many regions of the world groundwater is pumped from deeper
confined or leaky-confined aquifers (De Graaf et al., 2017). Under confined conditions,
groundwaters-streamflow interaction only occurs for the larger rivers that are deep enough to
penetrate the confining layer, while in leaky confined aquifers the interactions are more
complicated and delayed (Hunt, 2003).

There are many analytical solutions for calculating the stream depletion rate (SDR), defined
as the ratio of the volumetric rate of water abstraction from a stream to groundwater pumping
rate. These solutions differ in assumptions about the type of aquifer (unconfined, confined,
leaky-confined, multiple aquifers), stream bottom elevation, stream geometry and including
additional resistance from the streambed clogging layer or not. We refer to Huang et al.
(2018) for an extensive overview of solutions and when to apply them. These analytical
solutions typically involve a single well and a single stream, or, using apportionment
methods, a single well and stream networks (Zipper et al., 2019), while they consider streams
to be connected with the water table. Such analytical solutions could possibly be used for
multiple wells using e.g. superposition. However, for more complex situations, with multiple
wells, increasing withdrawal rates and streams changing from e.g. connected to disconnected,
numerical groundwater models need to be used. These have the disadvantage that they are
parameter-greedy, time-consuming to setup and often computationally expensive. Thus,
relatively simple analytical tools to assess the effects of extensive multi-well groundwater
pumping on groundwater and surface water systems at large are lacking.

Here, we introduce a simple analytical framework based on a lumped conceptual model of
aquifer-stream interaction under pumping. The framework aims to describe at larger scales,
i.e. large catchments and/or regional-scale phreatic aquifer systems, to what extent multi-well
groundwater withdrawal affects area-average groundwater heads and streamflow. It allows
for a shift in the nature of groundwater-surface water interaction, calculates at which critical
withdrawal rate such a shift is expected and when it is likely to occur after withdrawal
commences. It also provides estimates of streamflow depletion and the partitioning between
groundwater storage depletion and reduction in streamflow (capture). We envision that such
an analytical framework, when parameterized with parameters and inputs from a more
complex global-scale hydrological model, can be used as a screening tool for fast first-order
assessments of regional-scale groundwater sustainability, and for supporting hydroeconomic
models that require simple relationships between large-scale groundwater withdrawal rates
and the evolution of pumping costs and environmental externalities.

In the following, we first introduce the lumped conceptual model of large-scale groundwater
pumping with groundwater-surface water interaction. Next, we show its properties with an
extensive sensitivity analysis, followed with a global application of the model using inputs
and parameters from an existing global hydrological model (PCR-GLOBWB 2) and an
evaluation of its performance with estimates from in situ-observations, regional and global
groundwater models and satellites.

**2. Conceptual model of large-scale groundwater pumping with**
**groundwater-surface water interaction**

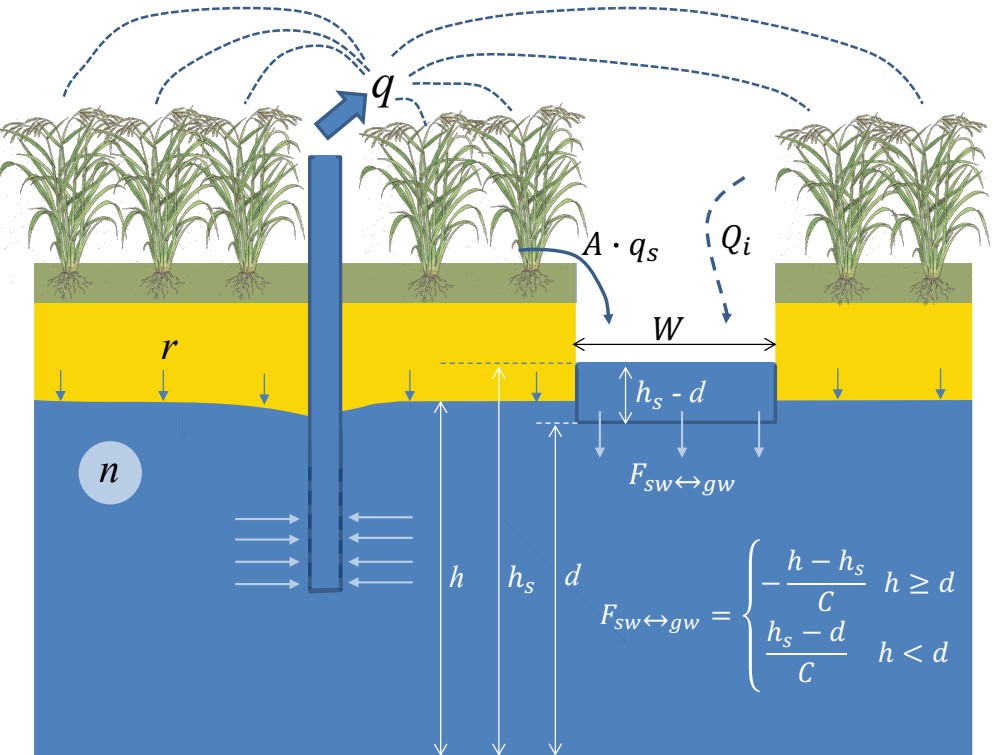

*Figure 2. Conceptual model of groundwater extracted (in this case for irrigation) from an*
*aquifer recharged by diffuse recharge and riverbed infiltration. Symbols are explained in the*
*text.*

A lumped conceptual hydrogeological model is proposed that allows for the analytical
treatment of area-average large-scale groundwater decline under varying pumping rates, yet
exhibits the properties of surface water-groundwater interaction. Consider a simplified model
of a phreatic aquifer subject to groundwater pumping (Figure 2). The volume of groundwater
pumped sums up all the pumping efforts of a large number of land owners that all draw water
from the same aquifer that can be seen as a common pool resource. Recharge consist of
diffuse recharge from precipitation and concentrated recharge from river-bed infiltration,
where river discharge comes from local surface runoff and from inflow from upstream areas
outside the area of interest.

Being of lumped nature, the model neglects (lateral) groundwater flow processes within the
aquifer and the mutual influence of multiple wells by treating the aquifer as one pool with a
given specific yield and unknown depth (i.e. physical limits are unknown) subject to pumping
treated as a diffuse sink. The latter is a simplification that represents the effects of hundreds
to thousands of wells of farmers spread more or less evenly across the aquifer. Also, we
assume withdrawal rate, surface runoff and river bed recharge to be constant in time,
neglecting seasonal variations that usually occur due to variation in crop water demand.
These simplifications allow us to represent the change of groundwater level $h$ with a simple
linear differential equation of the total aquifer mass balance:

$$n\frac{dh}{dt} = r + F_{gw \leftrightarrow sw}(h) - q \tag{1}$$

With
$h$: groundwater head (m)
$n$: specific yield (-)
$q$: pumping rate per area (m$^3$ m$^{-2}$ yr$^{-1}$)
$F_{gw \leftrightarrow sw}$: surface water infiltration (or drainage) flux density (m$^3$ m$^{-2}$ yr$^{-1}$)

The groundwater - surface water flux is modelled as follows:
$$F_{gw \leftrightarrow sw}(h) = \begin{cases} -\frac{h - h_s}{C} & h \geq d \\ \frac{h_s - d}{C} & h < d \end{cases} \tag{2}$$

with $h_s$ is the surface water level and $d$ the elevation of the bottom of the water course. The
parameter $C$ is a drainage resistance (yr) which pools together all the parameters of surface-
water groundwater interaction, i.e. the density or area fraction of surface waters, surface
water geometry and river/lake-bed conductance and the hydraulic conductivity of the aquifer.
Equation (2) is also used to describe groundwater-surface water interaction in numerical
groundwater such as MODFLOW (McDonald and Harbaugh, 2005), as well as in several
large-scale hydrological models (Döll et al., 2014; Sutanudjaja et al., 2018). This is a
simplification of the true interaction where in case of a detachment of the groundwater level
and the river bed ($h < d$) negative pressure heads can occur below the river bed and Equation
(2) may underestimate the river bed infiltration (Brunner et al., 2010). However, this latter
study also shows that errors remain within 5% in case the surface water is deep enough (> 1
m). Equation (2) provides a critical transition in terms of the effect of pumping on the
hydrological system. As long as the groundwater level is above the bottom of the surface
water network, the groundwater-surface water flux acts as a negative feedback on
groundwater level decline, at the expense of surface water decline. As the water table falls
below the bottom elevation (only possible if pumping rate $q$ is large enough; see hereafter),
surface water decline stops and progressive groundwater decline sets in.
The surface water level itself is a variable which is related to the surface water discharge $Q$
($m^3$ $y^{-1}$) and the groundwater level as follows:
$$Q = Wv(h_s - d) = Q_i + q_s A - F_{gw \leftrightarrow sw}(h)A \qquad (3)$$
with
$A$: The area over the (sub-)aquifer considered ($m^2$)
$q_s$: surface runoff (m $yr^{-1}$)
$Q_i$: influx of surface water from upstream ($m^3$ $yr^{-1}$)
$W$: Stream width (m)
$d$: Bottom elevation stream (m)
$v$: Stream flow velocity (m $yr^{-1}$)
The influx $Q_i$ is added to account for aquifers in dry climates where the surface water system
is fed by wetter upstream areas, e.g. mountain areas. The surface runoff $q_s$ (including shallow
subsurface storm runoff) also supplements the streamflow. Equation (3) lumps the
streamflow system overlying the phreatic aquifer system with a representative discharge,
water height, flow velocity and stream width taken constant in time. Equations (1)-(3)
together describe the coupled surface water-groundwater system where all parameters and
inputs remain constant with time and groundwater head $h$ and surface water levels $h_s$ change
over time as a result of groundwater pumping only.

In Appendix A expressions are derived for the following properties of the coupled system:

$q_{crit}$       Critical pumping rate ($m^3 \, m^{-2} \, yr^{-1}$) above which the groundwater level becomes

disconnected from the stream.

$t_{crit}$       Critical time (years after start of withdrawal) at which the groundwater level

becomes disconnected from the stream, i.e. $h < h_s$.

$h(t)$       Groundwater head (m) over time
$h(\infty)$    Equilibrium groundwater head (m) at $t=\infty$ that only occurs in case $q \leq q_{crit}$
$h_s(t)$     Surface water level (m) over time.
$h_s(\infty)$   Equilibrium surface water level (m), which is different when $q \leq q_{crit}$ than when

$q > q_{crit}$.

$Q(t)$      Surface water discharge ($m^3 \, yr^{-1}$) over time.
$Q(\infty)$    Equilibrium surface water discharge ($m^3 \, yr^{-1}$), which is different when $q \leq q_{crit}$

than when $q > q_{crit}$.

$q_{stor}(t)$   Part of the pumped groundwater that comes out of storage, which is different

when $q \leq q_{crit}$ than when $q > q_{crit}$.

$q_{cap}(t)$   Part of the pumped groundwater that comes from capture (reduction in

streamflow), which is different when $q \leq q_{crit}$ than when $q > q_{crit}$.


Table 1 provides an overview of the mathematical expressions derived for each of these
properties in Appendix A. The left column shows the stable regime where upon
commencement of pumping after some time an equilibrium is reached with equilibrium
groundwater levels $h(\infty)$, streamflow $Q(\infty)$ and surface water level $h_s$. The middle and right
columns show the results of unstable groundwater withdrawal. The behavior of $h(t)$, $Q(t)$ $h_s(t)$
follows that of the stable regime until time $t = t_{crit}$ when the groundwater level drops below
the bottom of the surface water. After this time the groundwater level $h(t)$ shows a persistent
decline and surface water level $h_s(t)$, streamflow $Q(t)$ and the fraction of water pumped from
capture become constant.







*Table 1. Overview of derived expressions for groundwater properties used in this paper*

| $\alpha = \dfrac{Q_i C + q_s AC + WvdC}{WvC + A}$   $\beta = \dfrac{A}{WvC + A}$   $q_{\text{crit}} = r + \dfrac{Q_i + q_s A}{WvC + A}$ | | |
|---|---|---|
| | $q > q_{\text{crit}}$ | |
| $q \le q_{\text{crit}}$ | $t_{\text{crit}} = \dfrac{nC}{1-\beta}\ln\left(\dfrac{qC}{qC - (rC+\alpha) + d(1-\beta)}\right)$ | |
| | $t \le t_{\text{crit}}\ (h \ge d)$ | $t > t_{\text{crit}}\ (h < d)$ |
| $h(t) = \dfrac{rC + \alpha}{1-\beta} - \left(\dfrac{qC}{1-\beta}\right)\left[1 - e^{-\left(\frac{1-\beta}{nC}\right)t}\right]$ <br><br> $h(\infty) = \dfrac{rC + \alpha - qC}{1-\beta}$ | $h(t) = \dfrac{rC + \alpha}{1-\beta} - \left(\dfrac{qC}{1-\beta}\right)\left[1 - e^{-\left(\frac{1-\beta}{nC}\right)t}\right]$ | $h(t) = d + \left[\dfrac{r-q}{n} + \dfrac{(Q_i + q_s A)}{n(WvC + A)}\right](t - t_{\text{crit}})$ |
| $h_s(t) = \alpha + \beta h(t)$ <br><br> $h_s(\infty) = \alpha + \dfrac{\beta(rC + \alpha - qC)}{1-\beta}$ | $h_s(t) = \alpha + \beta h(t)$ | $h_s = d + \dfrac{(Q_i + q_s A)C}{WvC + A}$ |
| $Q(t) = Q_i + q_s A - \dfrac{A\alpha}{C} + \dfrac{A(1-\beta)}{C}h(t)$ <br><br> $Q(\infty) = Q_i + (q_s + r - q)A$ | $Q(t) = Q_i + q_s A - \dfrac{A\alpha}{C} + \dfrac{A(1-\beta)}{C}h(t)$ | $Q = \dfrac{(Q_i + q_s A)WvC}{WvC + A}$ |
| $q_{stor} = qe^{-\left(\frac{1-\beta}{nC}\right)t}$ <br><br> $q_{cap} = q\left(1 - e^{-\left(\frac{1-\beta}{nC}\right)t}\right)$ | $q_{stor} = qe^{-\left(\frac{1-\beta}{nC}\right)t}$ <br><br> $q_{cap} = q\left(1 - e^{-\left(\frac{1-\beta}{nC}\right)t}\right)$ | $q_{stor} = q - \left(r + \dfrac{(Q_i + q_s A)}{(WvC + A)}\right)$ <br><br> $q_{cap} = r + \dfrac{(Q_i + q_s A)}{(WvC + A)}$ |



## 3. Local sensitivity analyses

Figure 3 shows the results of a sensitivity analysis for the critical withdrawal rate $q_{crit}$ and the critical time until the water table disconnects from the stream $t_{crit}$. For the stable regime ($q \le q_{crit}$) it shows the change in groundwater level at equilibrium $dh=h(0)-h(\infty)$, the change in streamflow at equilibrium $dQ = Q(0)-Q(\infty)$ and the *e*-folding time $t_{ef} = nC/(1 - \beta)$ of reaching the equilibrium after the commencement of pumping. For the unstable regime, we show the decline rate of the groundwater level $dh/dt$, the (constant) streamflow depletion $dQ$ and the constant fraction of capture ($f_{cap} = q_{cap}/q$). We stress that our sensitivity analysis is far from exhaustive (global) and that sensitivity plots are shown to provide a general feel of the behavior of the model and to show relationships between parameters and outputs that are of

interest to show. Unless they are varied on one of the axes, the parameter values used are the
reference values denoted in Table 2.

*Table 2. Reference parameter values used in sensitivity analyses.*

| Parameter | Value |
|---|---|
| Surface water system | |
| $A$ | 1000 km$^2$ |
| $q_s$ | 0.001 m d$^{-1}$ |
| $Q_i$ | 50 m$^3$ s$^{-1}$ |
| $d$ | 95 m |
| $W$ | 20 m |
| $v$ | 1 m s$^{-1}$ |
| Hydrogeology | |
| $C$ | 1000 d |
| $n$ | 0.3 |
| $r$ | 0.001 m d$^{-1}$ |


Figure 3a shows that the critical withdrawal rate increases with the relative abundance of
surface water due to upstream inflow and runoff and decreases with a decreased strength of
the surface water-groundwater interaction (increased value of $C$). For stable withdrawal rates
we see the largest equilibrium groundwater level declines with increased pumping rates and
decreased strength of surface water-groundwater interaction, i.e. decreased capture (Figure
3c). Figure 3e shows that the equilibrium reduction in streamflow to be proportional to
groundwater withdrawal rate as expected, but to depend only mildly on the upstream inflow.
The latter is caused by the two-way interaction between surface water and groundwater:
increasing inflow for a given withdrawal rate reduces groundwater level decline, which in
turn limits the loss of surface water to the groundwater. As follows from the expression
$t_{ef} = nC/(1 - \beta)$, the time to equilibrium (Fig. 3g) , i.e. the time until the pumped
groundwater originates completely from capture and no further storge changes occur, is
proportional to the resistance value $C$ and the specific yield, where the degree of
proportionality depends on the surface water properties. Figure 3g also shows that the time to
full capture can be very large, up to several decades.

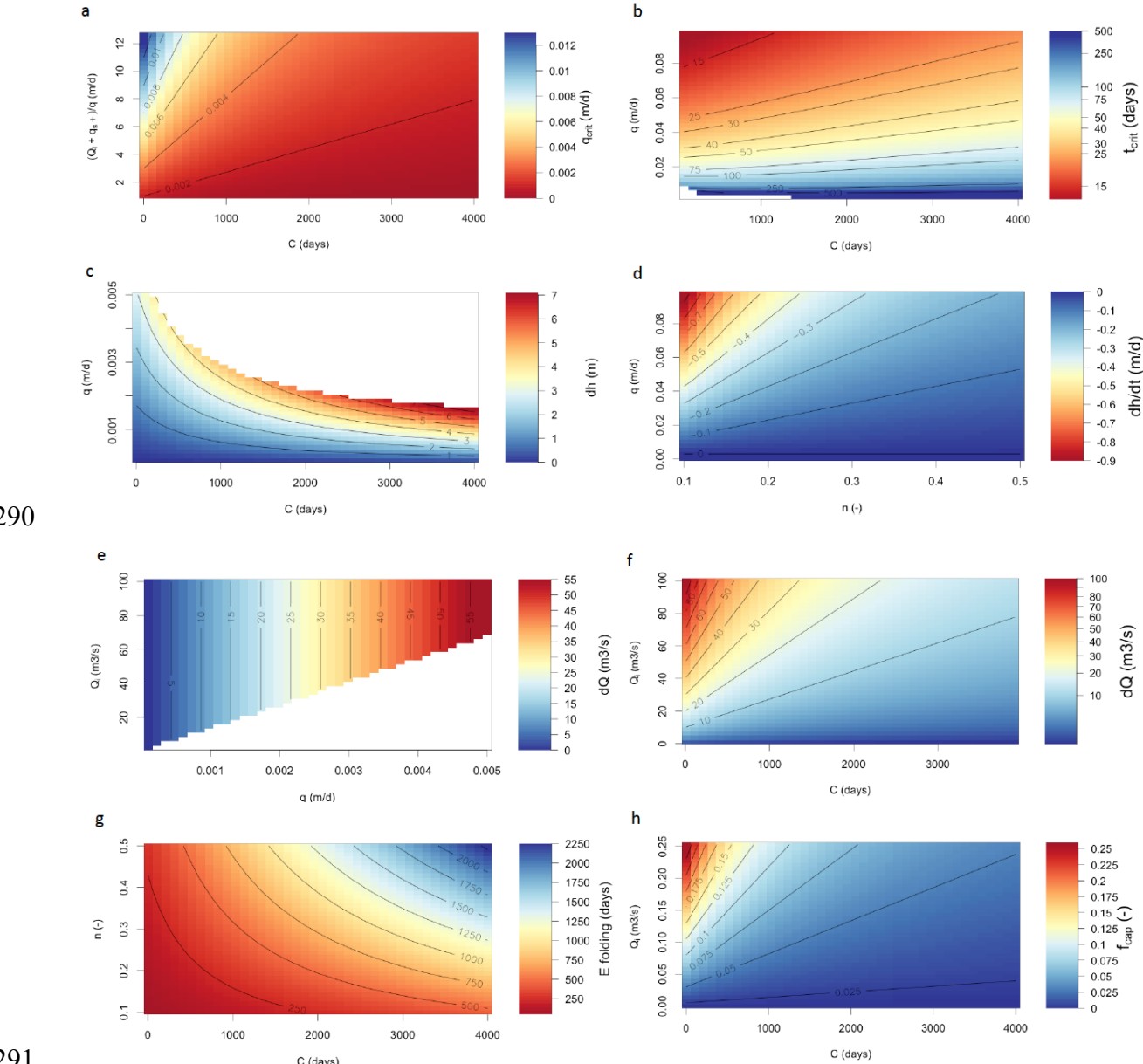

*Figure 3. Results of the sensitivity analyses showing parameter dependence of $q_{crit}$ (a) and $t_{crit}$ (b); variables under stable withdrawal: $dh=h(0)-h(\infty)$ (c), $dQ=Q(0)-Q(\infty)$ (e), $t_{ef} = nC/(1-\beta)$ (g)$t_{crit}$ and variables under unstable withdrawal and $t > t_{crit}$: $dh/dt$ (d), $dQ$ (f) and $f_{cap} = q_{cap}/q$.*

Figures 3b-h (right column) provides sensitivity plots of relevant variables in the unstable regime. Figure 3b shows that under the unstable regime, the time $t_{crit}$ to a transition from a connected to a non-connected groundwater table decreases with withdrawal rate, but slightly increases with $C$. The latter seems counter-intuitive at first, because a larger value of $C$ means reduced surface water contribution and therefore likely larger groundwater level decline rates and smaller values of $t_{crit}$. The equation for $h(t)$ in Table 1 (Equation A10 in the appendix) shows that this is indeed the case for early times but that for later times the decline rates are

reduced by a larger value of $C$ in the term factor $(1 - \beta)/nC$ in the exponential. Figures 3d-h
show sensitivity plots for $t > t_{crit}$ ($h < d$), i.e. groundwater levels are disconnected from the
surface water, groundwater is persistently taken out of storage and the capture becomes
constant. As expected, the groundwater level decline rates (Figure 3d) are proportional to
withdrawal rates and inversely proportional to specific yield. The final reduction in
streamflow (Figure 3f) for $t > t_{crit}$ decreases with the value of $C$ (limited surface water-
groundwater interaction), while the availability of surface water is important for smaller
values of $C$. Here, a larger inflow leads to larger losses because losses are proportional to the
surface water level which increases with inflow. Figure 3h resembles that of Figure 3f
because apart from the constant recharge, the fraction of capture is proportional to the
streamflow reduction which ends up in the pumped groundwater


## 4. Global-scale application

### *Global parameterization*

We applied the analytical framework to the global scale at 5 arc-minute resolution
(approximately 10 km at the equator) by obtaining parameters and inputs from the global
hydrology and water resources model PCR-GLOBWB 2 (Sutanudjaja et al., 2018, See Table
3 and Figures S1-S9 in the Supplement).  For the flux densities $q$, $q_s$, $r$, the discharge $Q_i$ and
the velocity $v$ we used the average values over the period 2000-2015. Note that for an
application of the analytical framework at a cell-by-cell basis, the reduction in streamflow $dQ$
in a given cell should be accounted for by reducing the inflow $Q_i$ to the downstream cell.
However, by using as inflow $Q_i$ the upstream discharge from a PCR-GLOBWB simulation
that includes human water use, upstream withdrawals from surface water and groundwater
are already accounted for. Note that they would also be implicitly included in case an
observation-based streamflow dataset (e.g., Barbarossa et al., 2019) would have been used for
$Q_i$. The groundwater-surface water interaction parameter $C$ is determined from the
characteristic response time $J$ of the groundwater reservoir in PCR-GLOBWB 2, which is
based on the drainage theory of Kraijenhoff-van de Leur (1958). From this solution and
Equation (2) it can be shown that $C=J/n$ (see Appendix B). Since the variables $q_{crit}$ and $t_{crit}$
depend heavily on the value of $C$ we have also included the dataset of groundwater response
time published by Cuthbert et al. (2019) to calculate the $C$ value.


*Table 3. Parameter and input values used in global-scale analyses at 5 arc-minute cells (~10*
*km ar the equator). All inputs obtained from PCR-GLOBWB 2 (Sutanudjaja et al., 2018),*
*except the C value obtained from PCR-GLOBWB and from Cuthbert et al. (2018); a input*
*variables averaged over the period 2000-2015.*

| Parameter | Value |
|---|---|
| Surface water system | |
| $A$ | Cell area 5 arc-minute cells ($m^2$) |
| $q_s$ | Sum of surface runoff and interflow (m d$^{-1}$) of a cell |
| $Q_i$ | Upstream discharge of a cell (m$^3$ d$^{-1}$) |
| $d$ | Stream elevation (m) based on bankfull discharge |
| $W$ | Stream width (m) based on bankfull discharge |
| $v$ | Calculated from bankfull discharge and stream depth (m d$^{-1}$) at Bankfull discharge, assuming $v$ to be dependent on terrain slope only. |
| Hydrogeology | |
| $C$ | $C = J/n$ (d), with $J$ the characteristic response time of the groundwater reservoir (Sutanudjaja et al. 2018) or groundwater response times from Cuthbert et al. (2019). |
| $n$ | Porosity values (-) from the groundwater reservoir in PCR-GLOBWB. |
| $r$ | Net recharge (recharge minus capillary rise) (m d$^{-1}$). |
| $q$ | Pumping rate (m d$^{-1}$). |


### Global results

Figure 4 shows the groundwater depletion rates $q$-$q_{crit}$ for the areas with unstable
groundwater withdrawal. The resulting patterns are similar to those calculated from previous
global studies (Wada et al., 2012: Döll et al., 2014) and show the well-known hotspots of the
world. Total depletion rates in Figure 4 are 158 km$^3$ yr$^{-1}$ (a) and 166 km$^3$ yr$^{-1}$ (b), which are in
the range of previous studies, e.g., 234 km$^3$ yr$^{-1}$ (Wada et al., 2012; year 2000), 171 km$^3$ yr$^{-1}$
(Sutanudjaja et al., 2018; 2000-2015) and 113 km$^3$ yr$^{-1}$ (Döll et al., 2014; 2000-2009).

The similarity of the groundwater depletion estimates with those obtained from global
hydrological models can be explained by the the fact that the way the groundwater-surface
water system is modelled in Figure 1 is similar to how the groundwater reservoirs and their
interaction with surface water have been implemented in global hydrological models such as
PCR-GLOBWB (De Graaf et al., 2015) and WGHM (Döll et al., 2014) (see also Appendix
B). Since the groundwater dynamics of latter models are (piece-wise) linear and groundwater
recharge in our model is applied directly in Equation (1) – i.e. the non-linear responses of the
soil system to precipitation and evaporation is bypassed -, forcing our model with average
fluxes *r, q, Qi* and $q_s$ and using the parameter *J* from PCR-GLOBWB yields almost the same
depletion rates as from the time varying model simulations with PCR-GLOBWB. The small
difference between our estimate (158 km³ yr⁻¹) and the value from PCR-GLOBWB 2
(Sutanudjaja et al., 2018) (171 km³ yr⁻¹) is explained by a resulting non-linearity not
accounted for: during dry periods some of the streams in the PCR-GLOBWB run dry and do
not contribute to the concentrated recharge flux. It should be noted that our results are
obtained at only a fraction of the computational costs of global hydrological models: a few
minutes at a single PC compared to 2 days on a 48-core machine with PCR-GLOBWB at 5
arc-minutes. Thus, the sensitivity to changing pumping rates or changes in recharge under
climate change can be quickly evaluated.

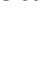

*Figure 4. Average groundwater depletion rates (q-q_crit) over 2000-2015 at 5 arc-minute*
*resolution calculated with the data from Table 2. (a) using C-values from Sutanudjaja et al.*
*(2018); (b) using C-values based on Cuthbert et al. (2019); (c) difference map a – b.*

Figure 5 shows the time to critical transition $t_{crit}$ from both datasets. It is quite striking that,
although the depletion rates are rather similar between datasets (Figure 4), the critical
transition times are much larger for the Cuthbert et al. (2018) dataset. These differences can
even add up to 2-3 orders of magnitude, which is extremely large. The reason is that the
characteristic response times based on Cuthbert et al. (2018) are much larger (also up to 2-3
orders of magnitude) than those based on PCR-GLOBWB. Since the e-folding time in the
stable regime is close to proportional to the $C$-value (e.g., Figure 3g), this is also true for the
critical transition time. The very large differences in response times between these two
datasets reveals that our method is only as good as its inputs and that critical transition times
and times to full capture calculated with our approach should be interpreted with care and as
order of magnitude estimates at best.

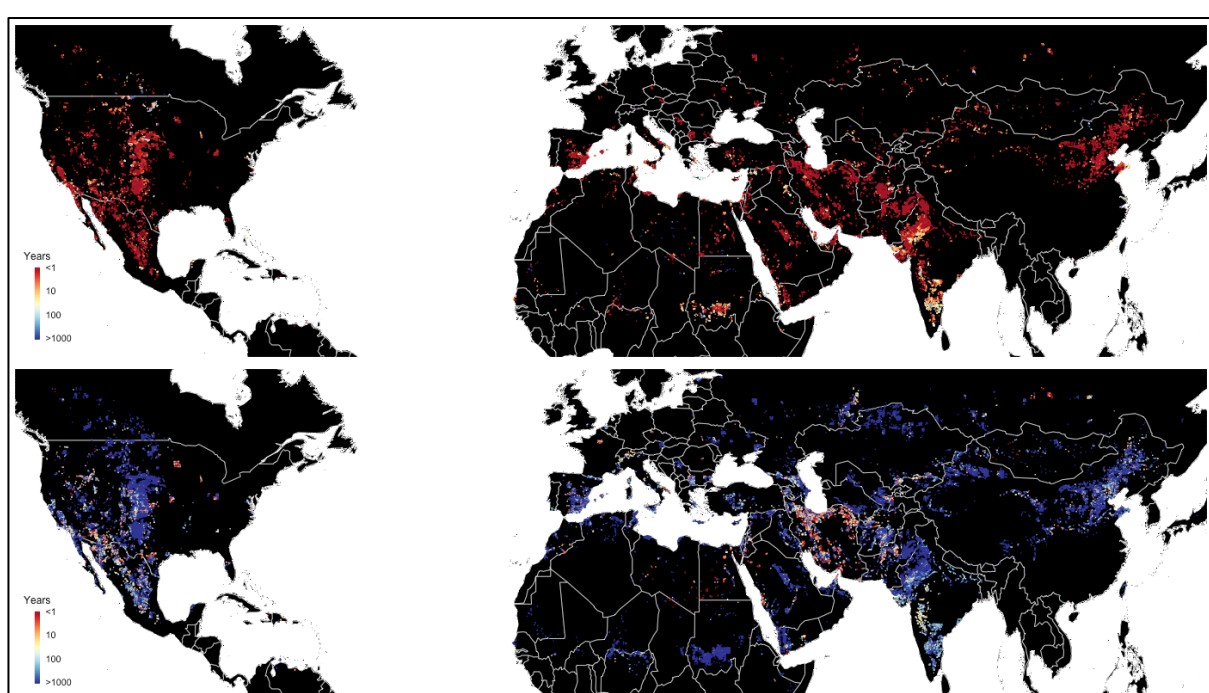

*Figure 5. Critical transition times (Critical time at which the groundwater level becomes*
*disconnected from the stream after start of pumping, i.e. $h < h_s$ in case $q > q_{crit}$) calculated*
*with the data from Table 1. The top figure uses C-values from Sutanudjaja et al. (2018) and*
*the lower figure from Cuthbert et al. (2019).*

To further explore the global impacts of groundwater withdrawal we calculated relevant
output variables for the areas that have been identified as subject to stable groundwater
withdrawal ($q \leq q_{crit}$; Figure 6) and unstable withdrawal ($q > q_{crit}$; Figure 7). Figure 6a
shows the equilibrium water table decline from stable groundwater withdrawal. We see the
largest declines occurring in areas with larger groundwater withdrawals, which are often
close to the depletion areas (Figure 4) and coincide with regions with limited surface water
occurrence due to a semi-arid climate (higher $C$-values). In contrast, the equilibrium decline
in streamflow (Figure 6b) is focused in areas with significant groundwater withdrawal and

higher surface water densities (low *C*-values), which are those areas that have a more semi-
humid climate where both groundwater and surface water use are present. These are also the
areas with relatively short times to equilibrium (Figure 6c).

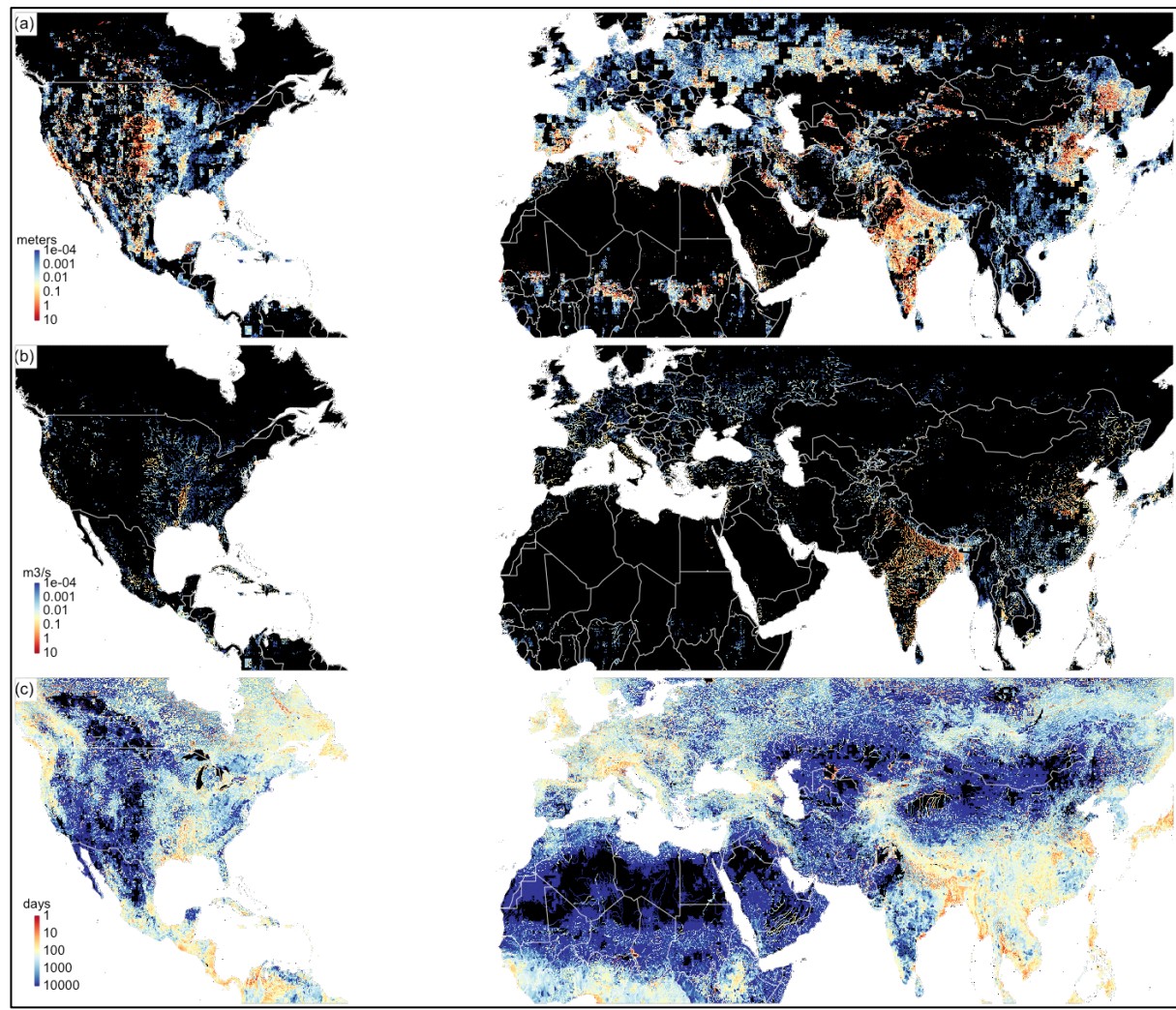

*Figure 6. Results for the areas with stable withdrawal rates ($q \leq q_{crit}$); (a) equilibrium
groundwater level decline (m); (b) equilibrium reduction of discharge ($m^3 \ s^{-1}$); (c) e-folding
time to complete capture (d); black areas are areas without groundwater withdrawal, with
unstable groundwater withdrawal or negligeable values ($< 10^{-4}$).*

As expected, the groundwater decline rates under unstable withdrawal (Figure 7a) mirror the
depletion rates (Figure 4). Estimates based on piezometers for major depleting areas are in
the order of 0.4-1.0 m yr$^{-1}$ in Southern California and the Southern High Plains aquifer
(Scanlon et al., 2012) and 0.1-1.0 m yr$^{-1}$ in the Gangetic plain (MacDonald et al., 2016). Our
estimates are in the lower end of those observed ranges, which could be partly explained by
the fact that, particularly in the U.S., groundwater withdrawal is from semi-confined aquifers,
leading to a larger head decline per volume out of storage than follows from the specific
yields used in our conceptual model. The largest change in streamflow and the highest

fraction of capture are found in areas where groundwater depletion coincides with the

presence of surface water, e.g. such as the Northern and Eastern part of the Ogallala aquifer,

the Indus basin and southern India.

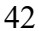

*Figure 7. Results for the areas with unstable withdrawal rates ($q > q_{crit}$); (a) groundwater level decline rate (mm yr$^{-1}$); (b) equilibrium reduction of discharge (m$^3$ s$^{-1}$); (c) fraction of capture (-); black areas are areas without groundwater withdrawal, with stable groundwater withdrawal or with negligeable values (< 10$^{-4}$).*

### *Sensitivity and evaluation of global results*

Critical parameters that determine the stream-aquifer interaction and hence many of the

outputs shown in Figures 4-7 are the stream-aquifer resistance parameter $C$ and the stream

bottom elevation $d$. We performed a local sensitivity analysis by changing the parameters $C$

and $d$ ±10% around their current values (Figures S3 and S6 in Supplement) and calculated the

relative change in the output per unit relative change in parameters $C$ and $d$. The results

(Supplementary Table S1) reveil that for most outputs the sensitivity to $C$ and $d$ is limited

(below unity). A notable exception is the sensitivity of $t_{crit}$ to $d$ which can be quite large,
particularly for the lower values of $C$ from Sutanudjaja et al. (2018). From the sensitivity
analysis we conclude that the global results are relatively robust to changes in the parameters
$C$ and $d$, except for the critical time to stream-aquifer disconnection which is sensitive to $d$
and to a lesser extend to $C$.

To evaluate our global results we compare these with observations and model results at
various scales, working from large to smaller scales (both in extent and resolution). These
include: aquifer average storage change from the Gravity Recovery and Climate Experiment
(GRACE) satellite, global-scale groundwater and streamflow depletion estimates from a
global groundwater model (De Graaf et al., 2019), continental-scale (conterminous U.S.)
groundwater and streamflow depletion estimates from Parflow-CLM (Condon and Maxwell,
2019) and groundwater flow and streamflow decline rates for the Republican River Basin
based on in-situ observations (Wen and Chen, 2006; McGuire, 2017).

From the results in Figure 4a (with $C$ from Sutanudjaja et al., 2018; assuming $q > q_{crit}$ and
$t > t_{crit}$) we computed average depletion rates of the world's major aquifers subject to
depletion (following Richey et al., 2015) and compared these with average trends in total
water storage (TWS) from GRACE (Gravity Recovery and Climate Experiment) gravity
anomalies over the period 2003-2015 (Figure 8). We used the JPL GRACE Mascon product
RL05M (Wiese, 2015; Watkins et al., 2015; Wiese et al., 2016). We did not correct TWS for
changes in other hydrological stores, assuming the latter to be approximately constant over a
13-year period in semi-arid areas with limited surface water and TWS trends to mainly reflect
groundwater depletion.  Figure 8 shows that the estimated depletion rates are reasonably
consistent with the GRACE estimates, particularly for the known hotspot aquifers with the
largest depletion. Notable exceptions are an overestimation of the depletion rate in the Paris
Basin and underestimation of depletion rates of the Maranhao Basin, the North Caucasus
Basin and the North African Aquifer Systems. These differences may be caused by errors in
withdrawal data from PCR-GLOBWB 2 (Supplementary Figure S9), errors in streamflow
leakage and errors that result from not correcting the GRACE products for possible secular
trends in other hydrological stores. A notable effect could be that by assuming aquifers to be
unconfined, we overestimate the leakage from surface water to groundwater in pumped
confined aquifers, leading to an underestimation of depletion rates. It should  also be noted
however that the aquifers whose depletion rates are underestimated have estimated GRACE

 trends between 1-10 mm yr[-1], just above the accuracy limit of GRACE TWS trends (viz.

Richey et al., 2015).

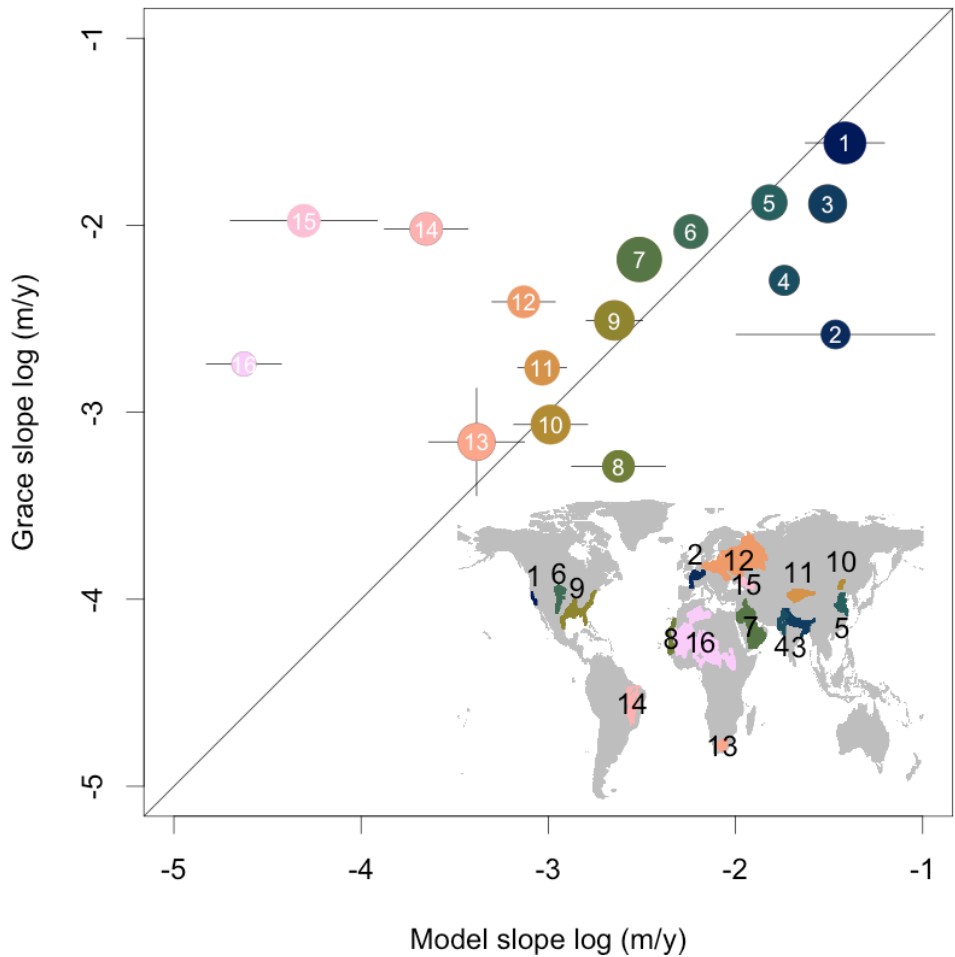

*Figure 8. Comparison of depletion rates in Figure 4a for major groundwater basins with*
*average depletion rates from GRACE (m yr[-1]).  Size of the circles is proportional to aquifer*
*area; crosses are standard errors in estimated mean aquifer trends; 1: Central Valley*
*(California); 2: Paris Basin; 3 Ganges-Brahmaputra Basin; 4; Indus Basin; 5: North China*
*Plane; 6. Ogallala (High Plains) Aquifer; 7. Arabian Aquifer System; 8: Senegalo-*
*Mauretanian Basin; 9: Atlantic and Gulf Coastal Plains Aquifer; 10: Song-Liao Basin; 11:*
*Tarim basin; 12; Russian Platform Basins; 13: Karoo Bason; 14: Maranhao Basin; 15:*
*North Caucasus Basin; 16: North African Aquifer Systems.*

At the global scale, we compared the head decline rate (mm d[-1]) calculated with the analytical
framework with average decline rates over the period 2000-2015 as obtained from the global
groundwater model of De Graaf et al (2019). Note that we restricted this comparison to the
areas with unstable withdrawal rates ($q > q_{crit}$, $t > t_{crit}$). The results shown in Figure S10 show
that the patterns of high and low values of the two estimates are similar, but that the
estimated decline rates from our analytical framework are larger than those estimated by De
Graaf et al. (2019). The most likely cause for the larger values in our approach is that it
neglects the impact of lateral flow (accross cell boundaries) or that the $J$-value of PCR-
GLOBWB used to calculate the $C$ parameter (see Appendix B) is too large so that leakage
from the streams is underestimated. Comparison of the stream depletion estimates from the
analytical framework (See Supplement Fig. S11; assuming $q > q_{crit}, t > t_{crit}$ or $q < q_{crit}, t >> t_{ef}$)
shows similar patterns to that of De Graaf et al. (2019), but also slightly larger values. Thus,
the most likely cause for the larger depletion values of our analytical framework (Figure S10)
is the neglect of lateral flow between cells.

At the continental scale, we compared groundwater storage changes (m) and stream depletion
(% of mean annual flow) across part of the conterminous U.S. obtained from a ParFlow-CLM
model (Condon and Maxwell, 2019) with the global estimates from our analytical
framework. ParFlow simulates coupled groundwater and surface water flow by solving the
3D Richards' equation and the diffusive wave equation respectively, while the community
land model CLM includes land surface processes such as evaporation, plant water use, snow
accumulation and snow melt. Condon and Maxwell (2019) calculate the total effects of
pumping from the predevelopment stage (1900 until 2008), while our global results are based
on the average withdrawal rates for the period 2000-2015.  To make our results comparable
with those of Condon and Maxwell (2019), we took their reported total storage loss of ~1000
km$^3$ since 1900 and determined the period length for which the total groundwater withdrawn
based on Sutanudjaja et al (2018) across the U.S. approximately equals 1000 km$^3$. This
resulted in the period 1965-2015. We subsequently recalculated the global maps using the
average groundwater withdrawal rate over 1965-2015 from Sutanudjaja et al. (2018).
The results are shown in the Supplementary Figures S12 (for $q > q_{crit}, t > t_{crit}$ ) and S13
($q > q_{crit}, t > t_{crit}$ or $q < q_{crit}, t >> t_{ef}$). Figure S12 shows again that the analytical approach
yields larger depletion estimates than ParFlow, but the results are more similar than with the
global model of De Graaf et al (2019). It is speculative at best to explain why the results of
Condon and Maxwell (2019) are more similar. One possible explanation may be that the
overestimation of decline rates due to ignoring lateral flow between cells in our approach is
partly offset by the neglect of headwater streams falling dry under continuous pumping. This
effect is included in ParFlow-CLM, which results in larger head decline rates that are closer
to ours. The global groundwater model of De Graaf et al (2019) does not include this effect
as streams in this model remain water carrying, even if the groundwater level drops below the
stream bottom elevation.
Figure S13 (top) shows the percentage reduction of streamflow by groundwater pumping
since predevelopment  as calculated by ParfFlow-CLM and Figure S13 (bottom) the
estimates based on the analytical framework. We show both maps for reference in the
Supplement, but it turns out that comparing the streamflow reduction of the analytical
framework with that of ParfFlow-CLM is inhibited by differences in model output and
presentation. The ParfFlow-CLM results represent cumulative $dQ$ as fraction of $Q$, whereas
the results from the analytical framework represent marginal $dQ$ as a fraction of $Q$, which
makes the results only comparable for the headwater catchments. Also, the difficulty of
comparison due to the resolution gap (ParfFlow-CLM: 1 km; analytical framework: 5
arcminutes ~ 10 km) is exacerbated due to the different map formats (vector and vs. raster).
Therefore, we refrain from further comments and show the maps as they are.

At the basin scale, we compared our global results with trends in groundwater head decline
and streamflow decline as obtained from observations of groundwater levels and surface
water discharge in the Republican River Basin (U.S.A.). The Republican Basin runs through
the northern part of the High Plains Aquifer system which is heavily influenced by
groundwater withdrawal. We used data from a study by Wen and Chen (2006) that estimated
trends in streamflow over the period 1950-2003 for 24 gauging stations spread across the
Republican river and its tributuaries. The trends were adjusted for possible trends in
precipitation and are therefore assumed to only reflect a decrease in streamflow as a result of
groundwater pumping. This resulted in 18 out of the 24 stations with significant negative
trends. Wen and Chen (2006) also provide groundwater level observations from three wells
with filters in the Ogallala formation at three location positioned in three representative
locations in the Republican Basin. We used the analytical framework with global parameters
(Table 3) but with the average values of $q$, $q_s$, $r$, $Q_i$ over the period 1960-2003 obtained from
PCR-GLOBWB (Sutatudjaja et al., 2009) to estimate at 5 arcminute resolution average
groundwater level decline rates (m yr$^{-1}$). Figure S14 in the Supplement shows box plots of
streamflow trends and groundwater head trends from the observations and from our
framework. The distribution of estimated streamflow decline overlaps with that from the
observed trends with a slight underestimation. The observed groundwater head decline rates
however are underestimated. This may be caused by the fact that we only have three
observations which are from a mostly confined aquifer where small storage coefficients lead
to larger decline rates.

To further investigate the performance of our method in reproducing groundwater level
declines at the sub-basin scale, we compare estimated groundwater level declines between
2002-2015 from 1522 groundwater wells in the Republican Basin obtained from McGuire,
(2017). Figure S15 shows maps and boxplots of observed groundwater level declines (m) and
declines estimated from the analytical framework. Although the overall pattern of
groundwater depletion in the Republican Basin is reproduced, there are occasional outliers in
the global estimates that are not seen in the observations. This is likely the result from the
global withdrawal data that are obtained by downscaling the total US groundwater
withdrawal to 5-arcminutes based on 5-arcminute estimates of total groundwater demand
(Sutanudjaja et al., 2018). Although these downscaled withdrawal rates are well verified at
the county-scale (See Wada et al., 2012), the mismatch at the 5-arcminute scale can be large.
Thus, when using global datasets, the analytical framework is limited to the sub-basin scale
and too coarse for local-scale estimates. Improvements can be expected when local data on
groundwater withdrawal are available at finer resolution.

*Critical limits to groundwater withdrawal for major basins*
We finish the result section by summarizing critical limits to groundwater withdrawal for the
major river basins of the world. In Figure 9a the median value of $q_{crit}$ is plotted for the major
basins in the world (sub-watershed level of HydroBasins, Lehner et al., 2008) together with
the areas where groundwater withdrawal is on average unstable over the years 2000-2015.
This figure provides, at first order, a global map of the maximum limit to physically stable
groundwater withdrawal rates. The parts of the world where the critical withdrawal rates are
very small largely coincide with the band of countries that experience high values of water
stress (Hofsté et al., 2019). This shows that there is little room in these areas to supplement
water demand without causing groundwater depletion.

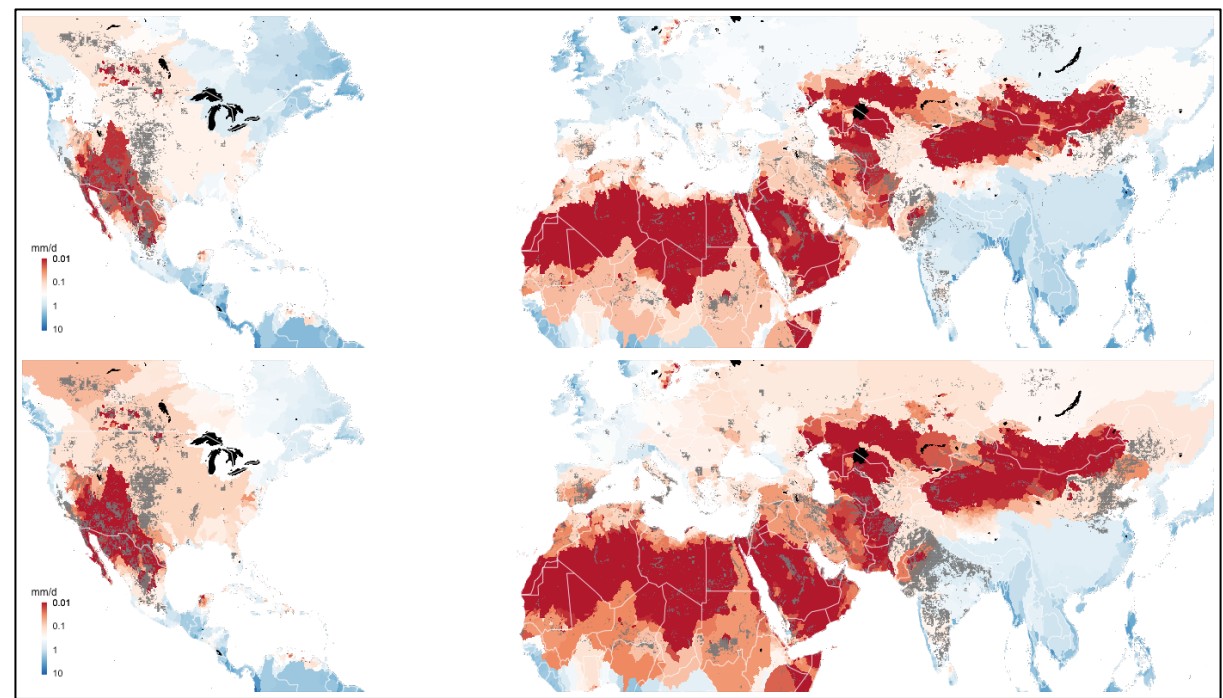


*Figure 9. Global limits to stable groundwater withdrawal rate; top: limit to physically stable*
*groundwater withdrawal mapped as the median $q_{crit}$ per sub-basin (based on Hydro-basins:*
*Lehner et al., 2008), grey-shaded areas are those for which $q>q_{crit}$; bottom: limit to*
*ecologically stable groundwater withdrawal mapped as the median $q_{eco}$ per sub-basin, grey-*
*shaded areas are those $q>q_{eco}$.*

The ecological limits to groundwater withdrawal, $q_{eco}$, can be defined as the withdrawal rate
that is low enough to prevent streamflow from dropping below some environmental flow
limit $Q_{env}$, i.e. a value that is high enough to safeguard the integrity of the aquatic ecosystems
(Linnansaari et al. 2013; Pastor et al 2014). The value of $q_{eco}$ can be calculated by inverting
Equation (A14) and taking $Q(\infty) < Q_{env}$:
$$q_{eco} = \frac{(Q_i + (q_s + r)A) - Q_{env}}{A} \qquad (4)$$

We note that environmental flows are usually defined during low flow conditions (Pastor et
al 2014; Gleeson and Richter, 2018), so it may be more appropriate to use the value of $Q(\infty)$
as the average over the summer half year instead of yearly averages. If we assume that the
average streamflow regime follows a cosine function with a period of 1 year, then the
average (natural) streamflow $Q_s$ in summer would be equal to:
$$Q_s = \left(1 - \frac{2}{\pi}\right)[(Q_i + (q_s + r)A] \qquad (5)$$

and $q_{eco}$ becomes:
$$q_{eco} = \frac{\left(1 - \frac{2}{\pi}\right)[Q_i + (q_s + r)A] - Q_{env}}{A} \qquad (6)$$

In Figure 9 (bottom) we have plotted $q_{eco}$ using, as an example, $Q_{env}$ to be 20% of the
average natural summer streamflow $Q_s$. The resulting map can be seen as a first order
approximation of the limits to ecologically stable groundwater withdrawal. In most cases,
$q_{eco} < q_{crit}$ as is also evident from the larger grey-shaded areas in the bottom figure compared
to the top figure. The results suggest that supplementing water demand by groundwater use in
the world's water stressed areas is limited under ecological constraints. We stress that the
sub-basin scale critical and environmental limits are meant for large-scale environmental
assessment, not for local groundwater management.

**4. Discussion and conclusions**

We have introduced an analytical framework based on a lumped conceptual model that
intents to describe to what extent groundwater withdrawal affects groundwater heads and
streamflow under changing regimes of groundwater-surface water interaction. By feeding the
framework with the parameters and inputs from a more complex hydrological model (i.e.,
PCR-GLOBWB), it can be used as a screening tool for regional-scale groundwater
sustainability. i.e., by providing a rich tableau of hydrologically and ecologically relevant
outputs at very limited computational costs. Another possible application is in
hydroeconomic modelling, where the equations in Table 1 can be used as regionally varying
hydrological response functions (Harou et al., 2009; MacEwan et al., 2017) in
hydroeconomic optimization – where model evaluations need to be fast - in order to infer
socially optimal pumping rates that include environmental externalities.

The estimated global groundwater and surface water depletion rates were compared with
observations and model results at various scales (support and extent), with mixed but overall
favourable results up to the sub-basin scale. Results show that the analytical framework
provides similar results to that of global hydrological models, but tends to overestimate the
groundwater depletion rates from groundwater flow models that account for lateral flow
between cells. Also, without calibration, the critical transient times, i.e, the time from
commencement of pumping till the detachment of the water table from the stream, as well as
the related time to full capture, are order-of-magnitude estimates at best. Finally, when using
global datasets, the analytical framework is limited to the sub-basin scale and too coarse for
local-scale estimates.

We stress that output variables that are related to critical environmental limits such as $q_{crit}$,
$q_{eco}$, $t_{crit}$ and $t_{ef}$ are difficult to validate directly, particularly at the larger scales at which our
framework operates. This would require large-scale pumping experiments or metering of
pumping wells in basins while surface water and groundwater are intensively monitored over
decades. As such, the critical limits are non-observables calculated with a model that is only
partly validated with a limited set of output variables, i.e. groundwater level decline and
streamflow depletion. We note however that this limitation is not restricted to our analytical
framework, but occurs for any analytical or numerical groundwater model used.

Clearly, many complicating factors are neglected in our approach, e.g.: underground spatial
heterogeneity, including the occurrence of multiple aquifer systems and semi-confined layers
that are present in many important alluvial groundwater basins; the variable depth and
topology of the surface water system and the intermittent nature of many streams in semi-arid
to semi-humid areas; and the locations of the wells with respect to the streams. Of these, the
neglect of confining layers may be one of the more crucial limitations of the approach. For
instance, a considerable part of the groundwater used for irrigation in the big alluvial basins
of the U.S. (e.g. Ogallala and Central Valley of California), where farmers have the financial
resources to drill deep wells (Perrone and Jasechko, 2019),  is pumped from deeper confined
aquifers. This means that the groundwater-surface water interaction is limited to the large
rivers and lakes only and that head decline per volume water pumped is larger than in
phreatic conditions. It would in principle be possible to include the effect of a confining layer
by using a larger value of the groundwater-surface water resistance parameter $C$, a smaller
value of recharge $r$ and a storage coefficient instead of specific yield. Similarly, the impacts
of seasonably variable boundary conditions of $q$, $q_s$ and $Q_i$ could be taken into account by
simple convolution, considering that the groundwater level responses $h(t)$ and d$h$/d$t$ (Table 1)
are respectively step and impulse responses of a linear system. Also, the effects of multiple
streams with variable stream bottom elevations could be included by extending the piecewise
linearization of Equation (2) to more domains (e.g. Bierkens and te Stroet, 2007). However,
we argue that such extensions are not in the spirit of the simple analytical framework
developed, which intents to provide first order sensitivities at larger scales. If the addition of
complexity is needed to provide more accurate assessments for a specific case, it would be
more logical to build a tailor-made numerical groundwater flow model.

We end with the note that a global application of our conceptual analytical framework is not
restricted to the use of data from the PCR-GLOBWB repository. The necessary fluxes $r, q, Q_i$
and $q_s$ can also be obtained from other repositories of multi-model re-analyses such as
EartH2Observe (Schellekens et al., 2017) and from the combination of remotely sensed
estimates of hydrological variables (Lettenmaier et al., 2015; McCabe et al., 2017), e.g.
estimating recharge and surface runoff from remotely sensed precipitation, evaporation and
soil moisture change, and using high-resolution global datasets on discharge (Barbarossa et
al., 2018) and river bed dimensions (Allen and Pavelsky, 2018; Lehner et al., 2018).

**Data availability.**

The data used in the global assessments provided by PCR-GLOBWB 2 can downloaded

from: https://doi.org/10.4121/uuid:e3ead32c-0c7d-4762-a781-744dbdd9a94b. The

groundwater response times of Cuthbert et al. (2019) can be found on:

https://doi.org/10.6084/m9.figshare.7393304 GRACE data used for validation are obtained

from: https://doi.org/10.5067/TEMSC-OCL05. The Republican River Basin well data from

2002-2015 can be downloaded from https://pubs.er.usgs.gov/publication/sim3373.

**Author contributions.**

MB conceived and designed the study. NW and MB performed the calculations. NW and

EHS performed the model validation. MB wrote the paper. All authors read, commented on,

and revised the manuscript.

**Competing interests**.

The authors declare that they have no conflict of interest.

**Acknowledgements.**

Niko Wanders acknowledges funding from NWO 016.Veni.181.049. The comments and

suggestions by the Editor, referee Grant Ferguson and four anonymous referees significantly

improved the manuscript.

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

## Appendix A: Conceptual model for regional-scale groundwater pumping with groundwater-surface water interaction

### A1. Basic equations

We repeat the three basic equations that make up the lumped conceptual model of regional-scale groundwater pumping with groundwater-surface water interaction:

The groundwater head as described with the total aquifer mass balance:

$$n\frac{dh}{dt} = r + F_{gw\leftrightarrow sw}(h) - q \tag{A1}$$

The groundwater - surface water flux:

$$F_{gw\leftrightarrow sw}(h) = \begin{cases} -\frac{h-h_s}{C} & h \geq d \\ \frac{h_s-d}{C} & h < d \end{cases} \tag{A2}$$

The surface water balance:

$$Q = Wv(h_s - d) = Q_i + q_s A - F_{gw\leftrightarrow sw}(h)A \tag{A3}$$

### A2. The case $h(t) \geq d$ and $q < q_{\text{crit}}$

We will start by analyzing the case that $h \geq d$, i.e. the groundwater level is attached to the surface water body. We further assume that $q < q_{\text{crit}}$, i.e. the groundwater withdrawal is such that the groundwater level never falls below the surface water bottom level $d$. In this case, the surface water flux $Q$ (m³/d) is related to the groundwater and surface water level as follows (See Figure A1):

$$Q = Wv(h_s - d) = Q_i + q_s A + \frac{h-h_s}{C}A \tag{A3}$$

with

$A$: The area over (sub-)aquifer considered (m²)

$q_s$: surface runoff (m yr⁻¹)

$Q_i$: influx of surface water from upstream (m³ yr⁻¹)

$W$: Stream width (m)

$d$: Bottom elevation stream (m)

$v$: Stream flow velocity (m yr⁻¹)

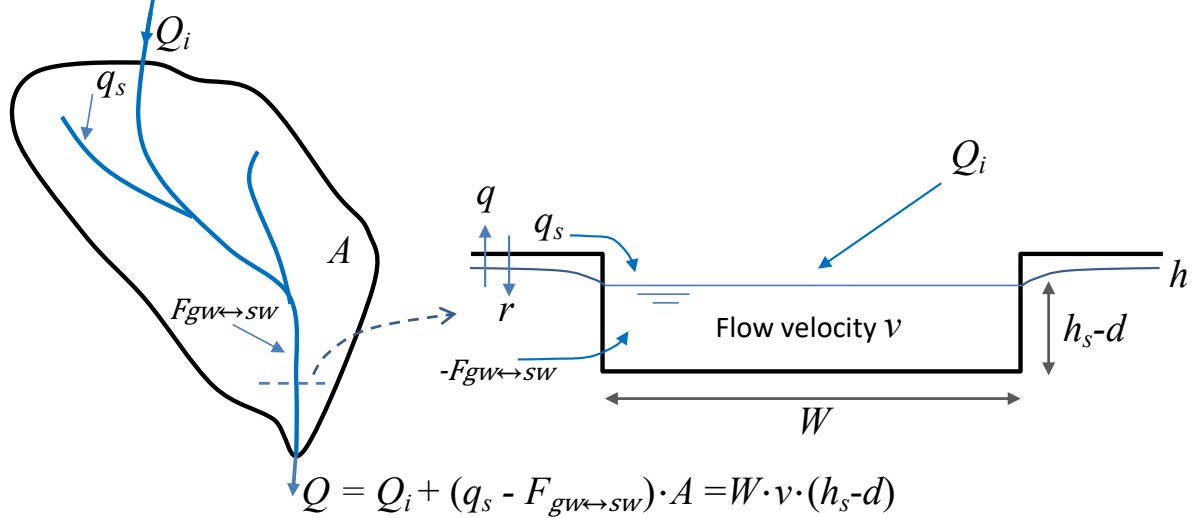

*Figure A1. Contributing fluxes to streamflow.*

Collecting $h_s$ on one side and the other terms on right side results in the following relation
between surface water height and groundwater head:

$$h_s(t) = \alpha + \beta h(t) \tag{A4}$$

with

$$\alpha = \frac{Q_i C + q_s A C + W v d C}{W v C + A} \tag{A5}$$

$$\beta = \frac{A}{W v C + A} \tag{A6}$$

From (A1) and (A2) the differential equation for groundwater level gives:

$$n \frac{dh}{dt} = r - \frac{h - h_s}{C} - q \tag{A7}$$

And after substituting (A4)

$$\Rightarrow n \frac{dh}{dt} = \left(r + \frac{\alpha}{C} - q\right) - \left(\frac{1 - \beta}{C}\right) h \tag{A8}$$

From (A8) follows the steady-state groundwater level under natural conditions ($q = 0$ and
d$h$/d$t$ =0):

$$\bar{h}_{nat} = \frac{r C + \alpha}{1 - \beta} \tag{A9}$$

Solving differential equation (A8) for initial condition (A9) then yields:

$$h(t) = \frac{r C + \alpha}{1 - \beta} - \left(\frac{q C}{1 - \beta}\right)\left[1 - e^{-\left(\frac{1-\beta}{n C}\right)t}\right] \tag{A10}$$

Which also gives the equilibrium groundwater level for $t \to \infty$:

$$h(\infty) = \frac{r C + \alpha - q C}{1 - \beta} \tag{A11}$$

The surface water level with time is given by (A4) and the final equilibrium surface water
follows from (A4) and (A11) as:
$$h_s(\infty) = \alpha + \frac{\beta(rc + \alpha - qC)}{1-\beta} \qquad (A12)$$
The surface water discharge as a function of time follows from combining (A3) and (A4):
$$Q(t) = Q_i + q_s A - \frac{A\alpha}{C} + \frac{A(1-\beta)}{C} h(t) \qquad (A13)$$
with $h(t)$ given by (A10). The equilibrium discharge is obtained by substituting (A11) for
$h(\infty)$ in (A13):
$$Q(\infty) = Q_i + (q_s + r - q)A \qquad (A14)$$
Which also follows logically from the water balance.

**A3. The critical withdrawal rate $q_{\text{crit}}$**
The critical withdrawal rate determines whether at larger times the water table drops below
the bottom of the surface and moves to the physically unstable regime. We seek $q$ such that
$h(\infty) = d$:
$$\frac{rC + \alpha - qC}{1-\beta} = d \qquad (A15)$$
From which follows:
$$q = \frac{rC + \alpha - d(1-\beta)}{C} \qquad (A16)$$
Substituting $\alpha$ and $\beta$ yields after some manipulation:
$$q_{\text{crit}} = r + \frac{Q_i + q_s A}{WvC + A} \qquad (A17)$$

**A4. Critical transition time $t_{\text{crit}}$ in case $q > q_{\text{crit}}$**
In case $q > q_{\text{crit}}$ at some time after pumping ($t_{\text{crit}}$) the groundwater level will fall below the
bottom elevation $d$ of the surface water. Before that time, it follows the water table decline
according to (A10). So, we can find $t_{\text{crit}}$ by solving it from:
$$h(t_{\text{crit}}) = \frac{rC + \alpha}{1-\beta} - \left(\frac{qC}{1-\beta}\right)\left[1 - e^{-\left(\frac{1-\beta}{nC}\right)t_{\text{crit}}}\right] = d \qquad (A18)$$
Solving an equation of the form $a - b[1 - e^{-cx}] = d$ gives as solution: $x = \frac{1}{c}\ln\left(\frac{b}{d-a+b}\right)$
from which follows from (A18):
$$t_{\text{crit}} = \frac{nC}{1-\beta}\ln\left(\frac{qC}{qC - (rC+\alpha) + d(1-\beta)}\right) \qquad (A19)$$

**A5. The case $q > q_{\text{crit}}$ and $t > t_{\text{crit}}$ ($h(t) < d$)**
In case the water table is below the bottom elevation of the stream, the water balance of the
stream reads (see Fig. A2):
$$Q = Wv(h_s - d) = Q_i + q_s A - \frac{h_s-d}{C}A \qquad (A20)$$
From which we can derive an equation for the minimum and constant elevation of the surface
water level (valid for $t > t_{\mathrm{crit}}$):
$$h_s = d + \frac{(Q_i+q_s A)C}{WvC+A} \qquad (A21)$$

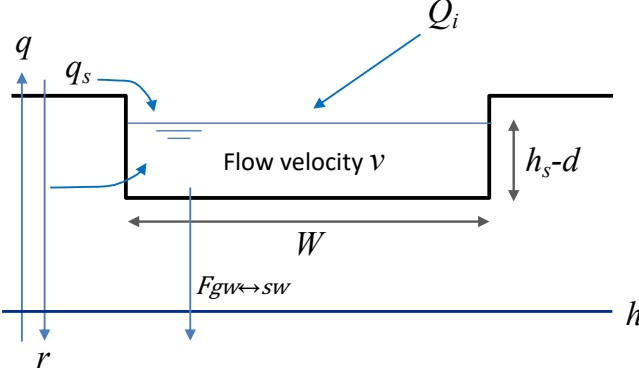


*Figure A2.  Water balance of a stream in case $q > q_{crit}$ and $t > t_{crit}$ ($h(t) < d$)*

The differential equation describing the change in groundwater with time now becomes:
$$n\frac{dh}{dt} = r - q + \frac{h_s-d}{C} \qquad (A22)$$
Substituting $h_s - d$ from (A21) then yields an equation for the groundwater decline rate:
$$\frac{dh}{dt} = \frac{r-q}{n} + \frac{(Q_i+q_s A)}{n(WvC+A)} \qquad (A23)$$
which is always negative since $q > q_{\mathrm{crit}}$. With initial condition $h(t_{\mathrm{crit}}) = d$ one obtains from
(A23) and equation for $h(t), t > t_{\mathrm{crit}}$:
$$h(t) = d + \left[\frac{r-q}{n} + \frac{(Q_i+q_s A)}{n(WvC+A)}\right](t - t_{\mathrm{crit}}) \qquad (A24)$$

**A5. Sources of pumped groundwater: $q < q_{crit}$ or $t < t_{crit}$ ($h(t) \geq d$)**
When neglecting direct evaporation from groundwater, the sources of pumped groundwater
in case $q < q_{crit}$ either come out of storage or from recharge that does not contribute to
streamflow. The latter is called "capture". From the water balance (A1) we thus find:
$$q = r + F_{gw \leftrightarrow sw}(h(t)) - n\frac{dh}{dt} \qquad (A25)$$
The first two terms constitute the water pumped from capture (with $F_{gw \leftrightarrow sw}$ negative in case
$h > h_s$ and positive when $h < h_s$) and the second term the water out of storage. Furthermore,
from differentiation of (A10) we have:
$$n\frac{dh}{dt} = -qe^{-\left(\frac{1-\beta}{nC}\right)t}$$ (A26)
Combining (A26) and (A25) then gives (since capture + out of storage add up to $q$):

$$q = \underbrace{q\left(1 - e^{-\left(\frac{1-\beta}{nC}\right)t}\right)}_{r + F_{gw\leftrightarrow sw}} + \underbrace{qe^{-\left(\frac{1-\beta}{nC}\right)t}}_{-n\frac{dh}{dt}}$$ (A27)


This shows that the fraction groundwater taken out of storage reduces over time until head
decline stops and all water comes out of capture.

**A6. Sources of pumped groundwater: $q > q_{crit}$ and $t > t_{crit}$ ($h(t) < d$)**
In case $q > q_{crit}$ and $t < t_{crit}$ the sources of pumped groundwater follow (A27). After the
groundwater table falls below the bottom elevation of the stream and $t > t_{crit}$ the sources of
water follow from (A23):
$$n\frac{dh}{dt} = r - q + \frac{(Q_i + q_s A)}{(WvC + A)}$$ (A28)
And therefore:
$$q = r + \frac{(Q_i + q_s A)}{(WvC + A)} - n\frac{dh}{dt}$$ (A29)
Since the third term is the storage change and capture plus storage change add up to $q$ we
have:
$$q = \underbrace{r + \frac{(Q_i + q_s A)}{(WvC + A)}}_{r + F_{gw\leftrightarrow sw}} + \underbrace{q - \left(r + \frac{(Q_i + q_s A)}{(WvC + A)}\right)}_{-n\frac{dh}{dt}}$$ (A30)


which shows that at after $t > t_{crit}$ the ratio of pumping from capture (i.e. recharge and surface
water leakage) and storage change becomes constant.


## Appendix B: Relationship between groundwater response time J and drainage resistance C

In PCR-GLOBWB 2 (Sutanudjaja et al., 2018) and in similar global hydrological models, the relationship between groundwater discharge $Q_g$ (m$^3$ m$^{-2}$ d$^{-1}$) and the volume $V_g$ (m$^3$/m$^2$) stored in the groundwater store is given by a simple linear relationship:

$$Q_g = \frac{V_g}{J} \tag{B1}$$

With $J$ the characteristic response time of the groundwater system (e-folding time of the recession) (yr). In some of the global models $J$ is obtained by calibration to low flows or recession curves. In PCR-GLOBWB it is calculated from transient drainage theory of Kraijenhoff-van de Leur (1958) as:

$$J = \frac{nL^2}{\pi^2 T} \tag{B2}$$

with $n$ the drainable porosity or specific yield, $L$ the average distance between water courses (derived from the drainage density per cell) and $T$ the aquifer transmissivity obtained from global hydrogeological datasets (e.g. Gleeson et al., 2014). A similar approach was used by Cuthbert et al. (2019) to derive groundwater response times.

The drainable volume of groundwater stored in the groundwater reservoir (m$^3$ m$^{-2}$) of a grid cell of a global hydrological model can also be expressed as: $V_g = n(h - h_s)$, with $h_s$ the surface water level and $h$ the groundwater level in the cell. Substituting this into (B1) we obtain the equivalent groundwater drainage equation for a grid cell:

$$Q_g = \frac{n(h - h_s).}{J} \tag{B3}$$

Comparing (B3) with (A2) shows that to obtain the same groundwater-surface water exchange in the global hydrological model and the conceptual analytical model we must have:

$$C = \frac{J}{n} \tag{B4}$$

Note that these relationships assume that the streams remain connected with the surface water, which is not entirely consistent with Equation A2.