# Peer review of "Large-scale sensitivities of groundwater and surface water to groundwater withdrawal"

_Hydrology and Earth System Sciences, 2020_

## Referee Comment (RC1) · Anonymous Referee #1 · 1 Feb 2021

—- Summary —- This study investigates the sensitivity of a linear reservoir-based model for groundwater pumping from unconfined aquifers and streamflow depletion, as well as applies this model for discretized cells of $\sim$100 km2 globally. Steady-state hydrologic parameters are input into the model with the outputs tracking the changing groundwater and surface water heads, presumably only considering a single model cell. A definition of sustainability is applied to the conceptual framework to explore the global spatial distribution of various model outputs and metrics.

—- Comments —-

1. I strongly disagree with the definition, and connected implications, of physical sustainability in this manuscript. According to the definition that groundwater pumping is sustainable so long as it doesn't cause the water table to disconnect from a surface waterbody is extreme. This definition means that nearly the entire flow of the river (i.e., Qi) could be extracted from the groundwater pumping and still be considered sustainable. A similar argument could be made that any streamflow lost due to groundwater pumping would be not sustainable, but the opposite is not necessarily the definition of a sustainable pumping regime (physical or otherwise). A dry or reduced flow river is not emblematic of sustainable abstractions in my opinion. Figure 1b can represent a physically unsustainable system, as the lost streamflow could lead to negative environmental effects downstream and could also cause feedbacks with downstream groundwater-surface water interactions. It would be fair to state that qcrit in this analysis is indicative of certain unsustainable hydrologic conditions, but it is not the threshold between sustainable and unstainable in either this conceptual framework or the real world. The opportunistic simplification of "capture" in this study is not complete, and the water budget and simplicity of the approach do not address capture in a sufficiently meaningful way to allow the application at the global scale to inform pumping management plans. As a somewhat connected note on this topic, the study does not need any definition or use of sustainability. If the study were instead posed on the potential disconnection of groundwater from surface water, then there would be no need for the value-loaded aspect of sustainability definitions. The "critical" outputs could be relabeled as "disconnection" or extreme flow reversal outputs. 2. What are the hydrologic restrictions of the constant hydrologic inputs? Importantly, it appears that the streamflow velocity remains constant while the depth and discharge can change. This suggests that the Q was not connected between cells, such that the pumping analysis was only providing information for each cell individually, such that Qi is constant and unaffected by pumping. This was not stated clearly in the text. This is important for then later calculations of depletion, comparisons with observational data (i.e., GRACE depletion rates), the delineation of "sustainable" vs "unsustainable" areas or watersheds, and the "global limit to sustainable gw pumping". These calculations represent nearly all of the location-specific results, and the lack of hydrologic connectivity is especially concerning for the calculation of qeco (Eq 4). 3. This study needs

to connect more clearly with the Zipper et al. (2019) paper rather than an offhanded statement on "a single well-network" method. This study is also applying a one well-one stream methodology that fits within the levels of complexity tested by Zipper et al. Treating the aquifer as an infinitely deep linear reservoir with uniform drawdown is less informative when applied to real locations (i.e., in the spatial analysis in this study) than the analytical approaches in Zipper et al. The distance a pumping well is from a stream is critical to calculating the streamflow and aquifer depletion, and the Zipper paper certainly serves as a foundation for global hydrologic studies that already have basically all of the information needed. Similarly, superposition was not mentioned in this study, but it could surely provide a very simple but powerful tool for calculating more realistic drawdowns. Forcing all drawdown across the model cell to be equal with this conceptualization also sets a very optimistic limit for what is being inappropriately labeled critical metrics for "sustainability". 4. Also of concern, relating to the Zipper paper, is the rather haphazard definition of the interaction term in this study, F. It includes the streambed conductance, a very difficult to constrain and important parameter, while also adding other geometries. If depletion is "often highly heterogeneous and incorrect estimates can lead to errors in estimated streamflow depletion (Fleckenstein et al., 2006; Irvine et al., 2012; Lackey et al., 2015)", as stated in Zipper et al., then I have a lot of trouble trusting the two versions of F (and J) used in this study, as neither sources were meant to provide such information on streambed connectivity to an aquifer. As such, the two sets of maps are pretty samples from an unknown distribution with unknown uncertainty. Also, the maps only show the actual values and never provide any information on the relative similarity/dissimilarity of the two calculations (other than being "striking", but not explained which is more realistic). Subtracting the two datasets and providing a map and histogram would give a sense of how important the unknown response time input for J and F is. These inherently include a length that may be inconsistent with the way this study was discretized. This again makes me question the utility of the many global outputs in this study 5. More information on the input datasets would be useful. For example, a description of the dataset used to apply "realistic" pumping rates for the

unconfined aquifers needs to be at least stated rather than requiring the reader to track it down elsewhere. The validity of these pumping rates sets the validity of all of the spatial results. Uncertainty in these pumping rates and resulting uncertainty in the results would also be useful, as the focus on mapped outputs implies the targeted impact of the global analysis is site-specific rather than global. 6. The connection between the PCRGLOB-WB (2) model needs to be stated in the beginning rather than in the discussion. The differences and novelty of this study needs to be presented at the beginning with the full context, rather than stating the similarity between this analysis and the previous modeling work "is not as surprising as it seems". The differences need to be VERY CLEARLY presented. Along this reasoning, the comparison of the depletion rates between this study and the former work needs to be more detailed. How many of the inputs between the models were different? How many of the equations? Are the integrated depletion rates for the globe smoothing over larger differences? 7. The comparison with GRACE data needs further development. How were the averages of depletion upscaled for these aquifers and some identification of the target areas would be useful? What are the unlabeled dots in Fig 5? What areas do they represent? What do the large misfits between the depletion rates, especially for the low rates from this study, indicate about the model performance and limitations? The issue of total water storage changes and an infinitely thick unconfined aquifer could be discussed in more detail. 8. The focus of the discussion of uncertainty on confining conditions is not all-encompassing, nor does it even assuage my concerns on the way the aquifer system was developed. Insufficient description of the various geometries and model inputs make it difficult to fully question the role of confined vs unconfined aquifers. An infinite depth unconfined aquifer system as the domain with an area the size of the grid cell is somewhat clear. Are the pumping rates only for the unconfined aquifer? If so, then why compare to GRACE TWS, as those are heavily tied to confined aquifer pumping in many areas? Justifications are lacking and explorations of the uncertainty of the effect of unconfined aquifer with infinite depth/storage on the results is missing from the analysis. 9. ". . .likely the simplest analytical form that can be devised" is amazingly

pompous and immediately false. Bragging at its finest. (Line 448). 10. The definition of F is different between Figure 2 and Equation 2. Reversing the inequality with a negative sign in Eq 2 results in problems. Figure 2 appears to be the correct definition, where negative F represents streamflow depletion and positive as baseflow. With Eq 2, h > hs leads to –F whereas h<hs leads to +F. In Eq 3, it appears that +F should lead to more streamflow, such that Fig 2 has the correct definition of F. A statement that +F is inflow into the surface water or something to that effect could help the reader follow this definition. Fig 2 should match the equations in the text and be consistent with the rest of the math. Similarly, some variables in Table 2 are capitalized when they are not in the text. 11. Numerous typos and misspellings throughout the paper. Lines 65, 68, 85, 101, 139, 149, 263, 283 (? or are tenths of years impressive?), 347, 388, 486, 717. 12. Ln 299 – inflow is flow in or out of the stream? Unclear here and elsewhere as this depends on perspective (towards surface water or towards groundwater?). 13. Ln 277 – Eq A30 mainly states that these fluxes negate each other, but the relationship of the ratio of these components is not known as q appears in this equation twice, unless additional assumptions are made (i.e., the ratio of the non-q components are equal to zero). 14. Ln 806 – distance, not difference 15. Ln 812 – it can also be set to other elevations, such as is implied in this study where pumpable groundwater exists below the streambed elevation. 16. All map figures are clipped to middle latitudes in the pdf I reviewed. I am unsure if this was intentional or not, but it seems arbitrary given the global extent of the analysis. 17. Separately on Qi, depending on the size of the watershed/catchment of interest, it seems strange to attribute the need for these to mountainous areas. Zero-order watersheds seem to also be depicted in Fig A1, which is absolutely not expected.

---

## Referee Comment (RC2) · GRANT FERGUSON (Referee) · 2 Feb 2021

Bridging the gap between global models that have used a water budget approach and more detailed numerical models at the regional scale by considering depletion and capture relationships is an important step forward. The approach used in this study holds promise in addressing the problem of the water budget myth (Bredehoeft, 2002) at the largest scales. However, there are some issues with how this work is framed in terms of sustainability and renewability along with some technical issues with the model.

The authors of this study cite papers that are inconsistent in how they define renewable groundwater resources. Bierkens and Wada (2019) use a mean residence time as a

measure of sustainability (line 50) while Wada (2016) uses a recharge-based approach (line 52). These definitions of renewability are problematic for multiple reasons. First, as pointed out by Bredehoeft (2002) and shown in the current study, renewal of groundwater is not restricted to background recharge but can also come from the reversal of hydraulic gradients at groundwater-stream interfaces once pumping begins. Second, mean residence time under background conditions is a function of flow system size is not connected to declines in water levels or streamflow in a simple way (Ferguson et al., 2020). Furthermore, when pumping a well to steady-state conditions (i.e. 100% capture) there is inevitably a portion of that groundwater that is non-renewable due to the cone of depression that develops to draw water towards the pumping well. The definitions of renewability used here are not useful in the context of groundwater management and are not necessary to support the ideas put forward in this manuscript. Removing discussion of these ideas will help to keep the focus on the problem of capture and depletion.

The definition of sustainability is problematic because of its specificity. Complete disconnection of water tables from streams as described in lines 87-92 is without a doubt problematic in humid and sub-humid areas but serious issues that would also be deemed unsustainable may occur before this happens, notably dry wells. This also creates issues with using groundwater in semi-arid and arid areas where losing and ephemeral streams exist, and groundwater flow systems exist on a larger scale than the 5 arc-minutes considered here. There are a variety of different conditions that need to be met to ensure sustainable development. Less rigid metrics for sustainable development of groundwater are likely more appropriate. The conditions put forth by Gleeson et al. (2020) that require maintaining water levels and flows above critical flow is vague but points to the need to understand disparate goals from various stakeholders and the unlikeliness of solving this problem with global models and one-size-fits-all metrics. As a community, we need to stop thinking in black and white in terms of sustainability. The authors can resolve this by focusing more explicitly on water tables and streams disconnect as an undesirable outcome rather than linking this disconnect to a

definition of sustainability.

The ability of the model to reproduce observed depletion rates is debatable because the time to full capture isn't properly considered in the model application. The simulation assumes that steady-state conditions existed before 2000 but depletion issues were known well before this time (Konikow, 2013). The match with GRACE (lines 339-350, Figure 5) data is coincidental because many regions should be on a later portion of the capture trajectory shown by Konikow and Leake (2014). Testing the model against observations would require more careful consideration of initial conditions and choice of simulated period. This may not be possible given the data available. However, presenting the simulation as an illustration of what would happen if pumping started in the year 2000 with no prior development is still a powerful demonstration of the capabilities of this model.

It is not surprising that this approach reproduces similar patterns to other global models of groundwater depletion (lines 315-320). The assumptions and approaches are not that different in the models mentioned. A comparison of the results presented here to large-scale numerical models may provide a better test of model performance. Condon and Maxwell's (2019) model examining the impacts of groundwater pumping on streamflow over a large section of the USA at a 1 km resolution provides such an opportunity. There are assuredly some differences in computation times but the numerical approach will likely be superior in resolving hydraulic gradients and could likely be done at a global scale in the near future. Furthermore, the analytical approaches reviewed by Zipper et al. (2019) are not restricted to single wells, as suggested in line 116. Invoking superposition with some of those concepts may provide another path forward to study capture and depletion at large scales. It is unclear that the approach used in the current study is "likely the simplest analytical form that can be devised to describe the effects of groundwater pumping at the larger scale" (lines 437-438). Objectively deciding the level of detail that effects of pumping need to be captured does not seem possible. Rather than making such claims, a more in-depth consideration

of how the global approach presented here compares to numerical models or analytical techniques at local and regional scales might provide important context for this work. Such a comparison may help to guide future efforts in advancing large-scale groundwater modelling.

This is a potentially important study in understanding large-scale groundwater depletion. While there are unresolved questions on the effectiveness of this approach is due to issues with initial conditions in the simulation, qualitatively it looks promising. The relationship between the results presented here and the threshold between sustainable and unsustainable development of groundwater is debatable. However, disconnection of water tables and streams is a clear indicator that groundwater pumping has resulted in an undesirable outcome and other thresholds may have already been passed.

References

Bierkens, M. F. and Wada, Y.: Non-renewable groundwater use and groundwater depletion: a review, Environmental Research Letters, 14(6), 063002, 2019.

Bredehoeft, J. D.: The water budget myth revisited: why hydrogeologists model, Groundwater, 40(4), 340–345, 2002.

Condon, L. E. and Maxwell, R. M.: Simulating the sensitivity of evapotranspiration and streamflow to large-scale groundwater depletion, Science advances, 5(6), eaav4574, 2019.

Ferguson, G., Cuthbert, M. O., Befus, K., Gleeson, T. and McIntosh, J. C.: Rethinking groundwater age, Nature Geoscience, 13(9), 592–594, 2020.

Gleeson, T., Cuthbert, M., Ferguson, G. and Perrone, D.: Global Groundwater Sustainability, Resources, and Systems in the Anthropocene, Annu. Rev. Earth Planet. Sci., https://doi.org/10.1146/annurev-earth-071719-055251, 2020.

Konikow, L. F.: Groundwater depletion in the United States (1900-2008), US Department of the Interior, US Geological Survey., 2013.

Konikow, L. F. and Leake, S. A.: Depletion and Capture: Revisiting "The Source of Water Derived from Wells," Groundwater, 52(S1), 100–111, 2014.

Wada, Y.: Modeling groundwater depletion at regional and global scales: Present state and future prospects, Surveys in Geophysics, 37(2), 419–451, 2016.

Zipper, S. C., Gleeson, T., Kerr, B., Howard, J. K., Rohde, M. M., Carah, J. and Zimmerman, J.: Rapid and accurate estimates of streamflow depletion caused by groundwater pumping using analytical depletion functions, Water Resources Research, 55(7), 5807–5829, 2019.

---

## Referee Comment (RC3) · Anonymous Referee #3 · 7 Feb 2021

Bierkens et al. present a simplified analytical methodology for first order approximation of the impacts of groundwater pumping on streamflow and ultimately groundwater sustainability. In addition to summarizing the methodology they provide global mappings of streamflow depletions and sustainable pumping limits. While I appreciate that the authors were very clear throughout the manuscript that this methodology is intended to be approximate, I still have very significant concerns and I don't feel that the manuscript in its current form has demonstrated that this approach is adequate to support the types of groundwater sustainability findings that are presented in figures 7-9 for the following reasons:

1. The approach presented here relies on a myriad of simplifying assumptions. While the authors do try to be very transparent in these assumptions, this does not make

them less concerning. Specifically, the steady state approach and the distributed well locations are big areas of concern in my opinion. For large scale aggregated analyses of declines this might be okay but for stream aquifer interactions well placement and timing is very important. The key advance of this paper is groundwater surface water interactions and therefore I think the bar is higher for some of these assumptions.

2. The authors present a sensitivity analysis for their approach which is a helpful illustration of the relationship between variables. However, for me this really only demonstrates that the general interactions are in the correct direction, which follows directly from the equations they used. Much more concerning to me is the uncertainty of the inputs to these equations at the spatial scales presented here and whether reliable estimates for some of the parameters can be generated at all. For example, how accurately can bed slope and bed elevation be captured at this resolution globally? How sensitive are the final results to the uncertainty in these values?

3. My biggest concern here is that the most important metrics that the authors are highlighting in their findings are not well validated. The authors present primarily comparisons to other global models which rely on similar assumptions and are working at similar spatial resolutions. It seems like it should be expected that the results here would be 'remarkably similar' (line 315-318). Before jumping to a global analysis I would like to see some rigorous evaluation of the model in some of the many heavily studied aquifers across the world comparing to regional models and observations. For example, observational groundwater ranges are reported for a few aquifers (Lines 379-382) but the authors only note that 'our estimates are in the lower end of those observed ranges' I think a much more quantitative comparison is need here.

4. Furthermore the validation that is provided here is really focused on groundwater depletions and I think the validation of the stream aquifer interactions or sustainable limits to groundwater withdrawals (the highlight of the paper) is lacking. If the main purpose of this work is to get to sustainability estimates and to connect to streamflow then these are the parts of the methodology which must be most thoroughly evaluated. I realize

that this information is not available globally (hence the novelty of this work). However, I don't see any reason why these behaviors cannot be rigorously and quantitatively evaluated in some example locations for which data or models are available.

5. Finally, the sustainability language in this paper is concerning to me. First of all because sustainability is a very subjective topic and it's not clear that the first order type approximations used here can really get at true sustainability. Second of all because I think these results can easily be misinterpreted based on how they are presented here. The authors do try to specify that this approach is only for first order approximations, but if that is the goal here then I think they should focus on using this methodology to provide ranges of potential groundwater depletions and stream interactions, and not be using this to present things link groundwater limits which can very easily be misinterpreted and miss-used.

Overall, I do think this is a well written paper that is clearly presented and easy to follow. Unfortunately, I am not convinced about the validity of the approach, and as a result the findings that are presented. I think a much more rigorous evaluation of the methodology is needed including quantitative analysis of every metric that is going to be presented in the findings. I completely understand that this methodology is intended to be approximate and will not perform as well as regional integrated models or intensive observational studies. However, I think these should still be the bar for comparison so that users of this approach can fully understand its strengths and weaknesses of the simplified approach, and so that any metrics that are too uncertain are not included.

---

## Author Comment (AC1) · 12 Feb 2021

**Reply to comments by anonymous reviewer #1**

*We thank reviewer 1 for his/her thorough review of our paper, which will help to improve the manuscript and the underlying study for the HESS readership. We will go over his/her comments point by point, with the comments in roman and our reply in italics. The specific actions we intend to perform in order to improve the paper are underlined.*

**Summary**
This study investigates the sensitivity of a linear reservoir-based model for groundwater pumping from unconfined aquifers and streamflow depletion, as well as applies this model for discretized cells of ~100 km2 globally. Steady-state hydrologic parameters are input into the model with the outputs tracking the changing groundwater and surface water heads, presumably only considering a single model cell. A definition of sustainability is applied to the conceptual framework to explore the global spatial distribution of various model outputs and metrics.

*Thanks for this summary that is mostly correct, except for the fact that it is stated that we apply this model to a single model cell. This is not correct. When we apply the framework globally, cells are connected in the sense that inflow in each cell depends on streamflow coming in from upstream cells.*

1. I strongly disagree with the definition, and connected implications, of physical sustainability in this manuscript. According to the definition that groundwater pumping is sustainable so long as it doesn't cause the water table to disconnect from a surface waterbody is extreme. This definition means that nearly the entire flow of the river (i.e., Qi) could be extracted from the groundwater pumping and still be considered sustainable. A similar argument could be made that any streamflow lost due to groundwater pumping would be not sustainable, but the opposite is not necessarily the definition of a sustainable pumping regime (physical or otherwise). A dry or reduced flow river is not emblematic of sustainable abstractions in my opinion. Figure 1b can represent a physically unsustainable system, as the lost streamflow could lead to negative environmental effects downstream and could also cause feedbacks with downstream groundwater-surface water interactions. It would be fair to state that qcrit in this analysis is indicative of certain unsustainable hydrologic conditions, but it is not the threshold between sustainable and unstainable in either this conceptual framework or the real world.

*The notion of sustainable groundwater withdrawal is indeed complex and highly debatable and its definition still in development, as shown in a recent review by Gleeson et al. (2020). We maintain that as long as streams are connected with the phreatic surface and pumping is such that it will not lead to a groundwater-stream disconnection, that this is a **physically** sustainable system, where an equilibrium water table will develop. This notion of physical sustainability is also used in the Gleeson et al. (2020) paper. We agree however, that this may still lead to damage to ecosystems or downstream effects on groundwater depth etc. Also, we agree that due to the simplicity of our lumped model approach (we will introduce the term lumped model in the abstract and introduction in a next version of the paper), we*

*do not account for the fact that disconnection may occur first higher up in the stream network and that disconnection is a spatiotemporally heterogenous process. We state this in our paper (lines 102-103). Regardless, the aim of this paper was not to pose an analytical framework for groundwater sustainability. Thus, the suggestion of the reviewer to delete references to sustainability and focus on the framework inspecting large-scale effects of groundwater withdrawal on surface water and groundwater including stream-aquifer disconnection is taken to heart. We will remove the term sustainability from the manuscript and refer to physically stable and unstable pumping regimes instead. The term stable then refers to pumping rates resulting in an equilibrium water table decline and instable pumping rates resulting in disconnection between groundwater and surface water and persistent decline of groundwater heads and groundwater depletion.*

> The opportunistic simplification of "capture" in this study is not complete, and the water budget and simplicity of the approach do not address capture in a sufficiently meaningful way to allow the application at the global scale to inform pumping management plans.

*Indeed, it is not complete. We do not take account of the impact of water table decline on evaporation and **diffuse** groundwater recharge. However, it is safe to say that the impact of water table decline on diffuse recharge is second order compared to the impacts on streamflow, in particularly in more semi-arid to semi-humid regions where soil saturation by shallow water tables is limited. Of course, our model does include the impact of water table decline on groundwater discharge to the stream and, in case $h < h_s$, recharge from the stream to the aquifer (**concentrated** recharge). So capture is taken into account in essence, certainly at the larger-scales that we state that our lumped model is said to operate on. We therefore, do not see any compelling argument why it cannot be applied at the global scale. Obviously, it will not inform pumping management plans for a single well or multiple wells at a local scale (we do not claim this), but may be informative for regional-scale effect studies of many wells.*

> As a somewhat connected note on this topic, the study does not need any definition or use of sustainability. If the study were instead posed on the potential disconnection of groundwater from surface water, then there would be no need for the value-loaded aspect of sustainability definitions. The "critical" outputs could be relabeled as "disconnection" or extreme flow reversal outputs.

*We agree with the reviewer (see our answer above).*

2.  What are the hydrologic restrictions of the constant hydrologic inputs? Importantly, it appears that the streamflow velocity remains constant while the depth and discharge can change. This suggests that the Q was not connected between cells, such that the pumping analysis was only providing information for each cell individually, such that $Q_i$ is constant and unaffected by pumping. This was not stated clearly in the text. This is important for then later calculations of depletion, comparisons with observational data (i.e., GRACE depletion rates), the delineation of "sustainable" vs "unsustainable" areas or watersheds, and the "global limit to sustainable gw pumping". These calculations

represent nearly all of the location-specific results, and the lack of hydrologic connectivity is especially concerning for the calculation of qeco (Eq 4).

*Streamflow velocity is indeed assumed constant. This is an assumption made to keep the relation between stream discharge Q and stream water elevation $h_s$ linear, resulting in linear ordinary differential equation to be solved. This assumption is further supported by the fact that streamflow between (larger size) rivers and streams and for the same streams/rivers over time is surprisingly constant, often varying between 0.5-1.5 m/s (see e.g., Figure 2. In Schulze et al., 2005). It is also based on the fact that for a rectangular channel it follows from Manning's equation that the derivative $dV/dh_s \sim h_s^{-1/3}$ which results in small changes in velocity with water depth for larger water depths (even more so for a trapezoidal channel). This does not mean however, that discharge in our model does not change as a result of groundwater pumping. It does! So, there is certainly a connection between pumping and streamflow and the impacts on environmental flow. The approach produces large-scale changes to downstream discharge due to groundwater pumping in an area given upstream inflow to this area.  What is not done in the global application is propagating the accumulated effects of pumping, i.e., by analyzing cell-by-cell following the large-scale streamflow network from upstream to downstream, although we could have been done this. Instead, in our global analysis, upstream withdrawals from surface water and groundwater are included as they come from PCR-GLOBWB. They would also be implicitly included in case an observation-based streamflow dataset (e.g., Barbarossa et al., 2019) would have been used. We will make this more clear when introducing the global application and discussing its connection with previous PCR-GLOBWB results.*

3. This study needs to connect more clearly with the Zipper et al. (2019) paper rather than an offhanded statement on "a single well-network" method. This study is also applying a one well one stream methodology that fits within the levels of complexity tested by Zipper et al. Treating the aquifer as an infinitely deep linear reservoir with uniform drawdown is less informative when applied to real locations (i.e., in the spatial analysis in this study) than the analytical approaches in Zipper et al. The distance a pumping well is from a stream is critical to calculating the streamflow and aquifer depletion, and the Zipper paper certainly serves as a foundation for global hydrologic studies that already have basically all of the information needed. Similarly, superposition was not mentioned in this study, but it could surely provide a very simple but powerful tool for calculating more realistic drawdowns. Forcing all drawdown across the model cell to be equal with this conceptualization also sets a very optimistic limit for what is being inappropriately labeled critical metrics for "sustainability".

*It was not our attention to supply an offhanded statement and dismiss the work of Zipper et al (2019). We certainly see its value and agree that it could be used by analyzing multiple wells by assuming superposition. We will acknowledge this in the next version of the paper. It remains however not possible to include the change in groundwater-surface water interaction from connected to disconnected in their approach. Our approach is indeed different in scale and less informative for local impacts. This manuscript does not present a single well single stream method. Instead, it is a lumped model (as opposed to a spatially explicit model of Zipper et al. (2019)) applicable to larger scales where all wells are lumped into a diffuse sink, assuming indeed the same drawdown. That this would lead to an*

*optimistic limit is not clear to us. It underestimates the effects of the wells that are relatively close to a water course and underestimates the effects of wells further away.*

4.  Also of concern, relating to the Zipper paper, is the rather haphazard definition of the interaction term in this study, F. It includes the streambed conductance, a very difficult to constrain and important parameter, while also adding other geometries. If depletion is "often highly heterogeneous and incorrect estimates can lead to errors in estimated streamflow depletion (Fleckenstein et al., 2006; Irvine et al., 2012; Lackey et al., 2015)", as stated in Zipper et al., then I have a lot of trouble trusting the two versions of F (and J) used in this study, as neither sources were meant to provide such information on streambed connectivity to an aquifer. As such, the two sets of maps are pretty samples from an unknown distribution with unknown uncertainty. Also, the maps only show the actual values and never provide any information on the relative similarity/dissimilarity of the two calculations (other than being "striking", but not explained which is more realistic). Subtracting the two datasets and providing a map and histogram would give a sense of how important the unknown response time input for J and F is. These inherently include a length that may be inconsistent with the way this study was discretized. This again makes me question the utility of the many global outputs in this study.

*Two points are made here, and we will answer them consecutively. First, the statement that the C or J value is haphazard. Well, it is not actually. In classical drainage theory, the flow geometry-related resistances and streambed resistance are often lumped into a single parameter called drainage resistance akin to our C-parameter (see e.g. Ernst (1956) and Kraijenhoff van de Leur (1958)) that can in fact be related to the domain geometry and hydraulic parameters and thus have a semi-physical basis. The drainage resistance parameter is also related to the characteristic response time of Cuthbert et al (2019). We therefore follow previous approaches of lumping groundwater flow. It is evident that water table and streamflow depletion decline due to pumping are sensitive to local heterogeneities. The fact that we use a lumped model does not mean that we negate the existence of such local heterogeneities. We do not resolve them because we aim to model large-scale (aquifer-scale, 100 km² grid cells) average responses to large-scale pumping. This is analogous to a lumped rainfall-runoff model of a catchment: using it does not mean that one denies that within a catchment heterogeneities of runoff response exist. Instead, it chooses to model a catchment total or average runoff response, often precisely because the local heterogeneities **cannot** be resolved. Also, the analytical depletion formulas used in e.g., Zipper et al. (2019) equally assume homogenous hydrogeology. Finally, streambed conductance is poorly constrained indeed, and this affects any groundwater modelling effort, both analytical as well as numerical.*

*The second issue is the maps comparing the depletion rates with the two datasets of C. It is a good idea to have a difference map between the PCR-GLOBWB C results and those obtained from Cuthbert et al (2019). We will do this in a next version of the paper.*

5.  More information on the input datasets would be useful. For example, a description of the dataset used to apply "realistic" pumping rates for the unconfined aquifers needs to be at least stated rather than requiring the reader to track it down elsewhere. The

validity of these pumping rates sets the validity of all of the spatial results. Uncertainty in these pumping rates and resulting uncertainty in the results would also be useful, as the focus on mapped outputs implies the targeted impact of the global analysis is site-specific rather than global.

*We thank the reviewer for pointing this out, and we are happy to comply. Apart from Table 3, we will provide a Supplementary Information file where we will provide maps of the input and parameter files used for the global analysis.*

6. The connection between the PCRGLOB-WB (2) model needs to be stated in the beginning rather than in the discussion. The differences and novelty of this study needs to be presented at the beginning with the full context, rather than stating the similarity between this analysis and the previous modeling work "is not as surprising as it seems". The differences need to be VERY CLEARLY presented. Along this reasoning, the comparison of the depletion rates between this study and the former work needs to be more detailed. How many of the inputs between the models were different? How many of the equations? Are the integrated depletion rates for the globe smoothing over larger differences?

*We respectfully disagree with the notion that the connection between PCR-GLOBWB needs to be stated up front. As we state in the discussion, a global application of the approach could also have been parameterized with the outputs of other global hydrological models or even global datasets based on observations and remote sensing. So, apart from us authors also being responsible for building and maintaining PCR-GLOBWB, there is no intended connection.*

*What is true is that the stream-aquifer interaction equation (Equation 2) is similar to what is used in PCR-GLOBWB, but also in other global hydrological models such as WGHM and even in the parameterization of the river package of MODFLOW. This is exactly what we state in Chapter 2 right at the beginning. Otherwise, PCR-GLOBWB is very different. It does not use any of the analytical solutions shown in Table 1, but rather uses a spatio-temporal discrete approach (time explicit) to solve the water balance equations. The analytical expressions are based on time-invariant forcing of the system and thus simplified. Still, they provide similar results close to instantaneously, instead of after days of numerical integration. In hindsight, this similarity can indeed be explained by the linearity of the groundwater reservoir that is also present in PCR-GLOBWB. We agree that this discussion is best done earlier and we will move the discussion about the similarity in results between our model and PCR-GLOBWB to the results section. We will also provide a pixel-by-pixel difference map with PCR-GLOBWB depletion in the Supplementary Information to add more detail and additionally with depletion rates from a global groundwater model (De Graaf et al., 2019) (also upon a request of Reviewer #2).*

7. The comparison with GRACE data needs further development. How were the averages of depletion upscaled for these aquifers and some identification of the target areas would be useful? What are the unlabeled dots in Fig 5? What areas do they represent? What do the large misfits between the depletion rates, especially for the low rates from this study, indicate about the model performance and limitations? The issue of total

water storage changes and an infinitely thick unconfined aquifer could be discussed in more detail.

*We thank the reviewer for these suggestions. We will add a map with shape files of the aquifer systems identified in the scatter plot to the Supplementary Information. We will also identify the unexplained dots and add more information on how the average depletion rates were calculated. We stress that the thickness of the aquifer is not an issue in our lumped conceptual model. We only present rate of storage change and do not presume to make predictions of when an aquifer becomes depleted without knowledge and inclusion of aquifer thickness or maximum pumping depth.*

8. The focus of the discussion of uncertainty on confining conditions is not allencompassing, nor does it even assuage my concerns on the way the aquifer system was developed. Insufficient description of the various geometries and model inputs make it difficult to fully question the role of confined vs unconfined aquifers. An infinite depth unconfined aquifer system as the domain with an area the size of the grid cell is somewhat clear. Are the pumping rates only for the unconfined aquifer? If so, then why compare to GRACE TWS, as those are heavily tied to confined aquifer pumping in many areas? Justifications are lacking and explorations of the uncertainty of the effect of unconfined aquifer with infinite depth/storage on the results is missing from the analysis.

*We acknowledge that we do not take into account that many aquifers are confined. Ignoring that an aquifer is actually confined, like we do, would have a big effect on groundwater-surface water interactions and would likely underestimate storage decline. Still, we can compare with GRACE to see how "wrong" we are. We will extend the discussion around the possible effects of ignoring confined aquifers when discussing Figure 5 about the comparison with GRACE.*

9. ". . .likely the simplest analytical form that can be devised" is amazingly pompous and immediately false. Bragging at its finest. (Line 448).

*This is not a very courteous way of saying that we overstate our case. We stand corrected and will remove the sentence.*

10. The definition of F is different between Figure 2 and Equation 2. Reversing the inequality with a negative sign in Eq 2 results in problems. Figure 2 appears to be the correct definition, where negative F represents streamflow depletion and positive as baseflow. With Eq 2, h > hs leads to –F whereas hh > hs leads to –F whereas h < hs leads +F. In Eq 3, it appears that +F should lead to more streamflow, such that Fig 2 has the correct definition of F. A statement that +F is inflow into the surface water or something to that effect could help the reader follow this definition. Fig 2 should match the equations in the text and be consistent with the rest of the math. Similarly, some variables in Table 2 are capitalized when they are not in the text.

*We thank the reviewer for noticing this inconsistency. We have aligned the sign of F in Figure 2 with the equations in the text. If F is positive it contributes to groundwater (depletes*

*streamflow) and when negative to streamflow (groundwater discharge). We have added a
sentence to this effect to the text. We have also corrected the inconsistency in low-upper
case between Table 2 and the text.*

11. Numerous typos and misspellings throughout the paper. Lines 65, 68, 85, 101, 139, 149,
263, 283 (? or are tenths of years impressive?), 347, 388, 486, 717. 12. Ln 299 – inflow is
flow in or out of the stream? Unclear here and elsewhere as this depends on perspective
(towards surface water or towards groundwater?).

*We thank the reviewer for noticing. We have corrected the typos. "tenths of years" is
"Dunglish" for "decades". We will also better clarify what inflow means.*

11. Ln 277 – Eq A30 mainly states that these fluxes negate each other, but the relationship
    of the ratio of these components is not known as q appears in this equation twice,
    unless additional assumptions are made (i.e., the ratio of the non-q components are
    equal to zero).

*We don't think so. The capture part can be calculated (which is actually $q_{crit}$), which is always
smaller than the pumping rate q in case of $q > q_{crit}$. Once that is known, the remaining part
comes out of storage, which also follows directly from the groundwater decline rate (A24).
The ratio can be calculated if pumping rate q is known.*

13. Ln 806 – distance, not difference

*Thank you for noticing. Corrected.*

14. Ln 812 – it can also be set to other elevations, such as is implied in this study where
pumpable groundwater exists below the streambed elevation.

*This is a correct observation. In that case the analogy with the Kraijenhoff van de Leur (1958)
solution breaks down because the latter does not account for disconnected streams. We do
assume that J remains the same though. We will state this assumption in the revised paper.*

15. All map figures are clipped to middle latitudes in the pdf I reviewed. I am unsure if this
was intentional or not, but it seems arbitrary given the global extent of the analysis.

*This clipping was done intentionally. The reason is that all the major groundwater pumping
and depletion occurs between 60º north and 60º south. This allows us to show the major
features while saving space. All global numbers are based on integrating across the entire
globe however.*

16. Separately on Qi, depending on the size of the watershed/catchment of interest, it
seems strange to attribute the need for these to mountainous areas. Zero-order watersheds
seem to also be depicted in Fig A1, which is absolutely not expected.

*Figure A1 is just a schematic and the tributaries do not represent first order catchments at
the scale of a lower river basin. Mentioning mountainous areas as source of inflow comes*

*from the fact that mountain front recharge is an important source of recharge in many of the heavily irrigated semiarid regions of the world.*

**References**

Barbarossa, V., Huijbregts, M., Beusen, A. et al. (2018). FLO1K, global maps of mean, maximum and minimum annual streamflow at 1 km resolution from 1960 through 2015. *Sci Data* 5, 180052.

Cuthbert MO, Gleeson T, Moosdorf N, Befus KM, Schneider A, et al. (2019). Global patterns and dynamics of climate-groundwater interactions. *Nat. Clim. Change* 9:137–41

de Graaf, I.E.M., Gleeson, T., van Beek, L.P.H., Sutanudjaja, E.H. and Bierkens, M.F.P. (2019). Environmental flow limits to global groundwater pumping. *Nature* 574, 90-108.

Ernst, L.F. (1956). Calculation of the steady flow of groundwater in vertical cross-sections. *Neth. J. of Agric. Sci*. 4, 126–131.

Gleeson,T., Cuthbert, M.O.,  Ferguson, F. and Perrone (2020). Global groundwater sustainability, resources, and systems in the Anthropocene. *Annu. Rev. Earth Planet. Sci.* 2020. 48:17.1–17.33

Kraijenhoff van de Leur, D. A. (1958). A study of non-steady ground-water flow with special reference to the reservoir-coefficient, *De Ingenieur* 19, 87–94.

Schulze, K., Hunger, M and Doll, P. (2006).  Simulating river flow velocity on global scale. *Adv. Geosci*. 5, 133–136.

---

## Author Comment (AC2) · 12 Feb 2021

**Reply to comments by reviewer Grant Ferguson.**

*We thank Grant Ferguson for his thorough review of our paper and his insights, which will help to improve the manuscript and the underlying study for the HESS readership. We will go over his comments point by point, with the comments in roman and our reply in italics.* *The specific actions we intend to perform in order to improve the paper are underlined.*

Bridging the gap between global models that have used a water budget approach and more detailed numerical models at the regional scale by considering depletion and capture relationships is an important step forward. The approach used in this study holds promise in addressing the problem of the water budget myth (Bredehoeft, 2002) at the largest scales.

*We thank the review for his encouraging words.*

However, there are some issues with how this work is framed in terms of sustainability and renewability along with some technical issues with the model. The authors of this study cite papers that are inconsistent in how they define renewable groundwater resources. Bierkens and Wada (2019) use a mean residence time as a measure of sustainability (line 50) while Wada (2016) uses a recharge-based approach (line 52). These definitions of renewability are problematic for multiple reasons. First, as pointed out by Bredehoeft (2002) and shown in the current study, renewal of groundwater is not restricted to background recharge but can also come from the reversal of hydraulic gradients at groundwater-stream interfaces once pumping begins. Second, mean residence time under background conditions is a function of flow system size is not connected to declines in water levels or streamflow in a simple way (Ferguson et al., 2020). Furthermore, when pumping a well to steady-state conditions (i.e. 100% capture) there is inevitably a portion of that groundwater that is non-renewable due to the cone of depression that develops to draw water towards the pumping well. The definitions of renewability used here are not useful in the context of groundwater management and are not necessary to support the ideas put forward in this manuscript. Removing discussion of these ideas will help to keep the focus on the problem of capture and depletion.

*The reviewer refers to an important point in that renewability in terms of recharge or mean renewal time may be debatable, as indicated by one of his recent commentaries in nature Geoscience (Ferguson et al. (2020). We do not want to engage in a debate about the proper definition of non-renewable groundwater use here, as it is not necessary for this paper as the reviewer rightfully states. The introduction was to make the point that groundwater overuse leads to groundwater depletion, which may be seen as non-renewable groundwater use if it takes a long time of the water taken out of storage to recover (viz. recovery times as suggested by the Ferguson and others in their papers).* *Therefore, we have changed the phrasing the introduction* *to: "This has greatly intensified the dependence of irrigated crops on groundwater withdrawal (Wada et al., 2012) and caused a steady increase of groundwater depletion rates (Wada and Bierkens, 2019). Recent estimates of current groundwater withdrawal range approximately between 600-1000 km$^3$ yr$^{-1}$ leading to estimated depletion rates of 150-400 km$^3$ yr$^{-1}$ (Wada, 2016)."*

The definition of sustainability is problematic because of its specificity. Complete disconnection of water tables from streams as described in lines 87-92 is without a doubt problematic in humid and sub-humid areas but serious issues that would also be deemed unsustainable may occur before this happens, notably dry wells. This also creates issues with using groundwater in semi-arid and arid areas where losing and ephemeral streams exist, and groundwater flow systems exist on a larger scale than the 5 arc-minutes considered here. There are a variety of different conditions that need to be met to ensure sustainable development. Less rigid metrics for sustainable development of groundwater are likely more appropriate. The conditions put forth by Gleeson et al. (2020) that require maintaining water levels and flows above critical flow is vague but points to the need to understand disparate goals from various stakeholders and the unlikeliness of solving this problem with global models and one-size-fits-all metrics. As a community, we need to stop thinking in black and white in terms of sustainability. The authors can resolve this by focusing more explicitly on water tables and streams disconnect as an undesirable outcome rather than linking this disconnect to a definition of sustainability.

*We concur. Reviewer #1 made the same objections against our use of the term physical sustainability. Therefore, we will not use the term in the next version of the paper and focus on stream-groundwater disconnection. We do however need to use a term to distinguish between the two regimes to avoid lengthy descriptions and have decided to use "stable" ($q <= q_{crit}$) and unstable ($q > q_{crit}$) withdrawal regimes.*

The ability of the model to reproduce observed depletion rates is debatable because the time to full capture isn't properly considered in the model application. The simulation assumes that steady-state conditions existed before 2000 but depletion issues were known well before this time (Konikow, 2013). The match with GRACE (lines 339- 350, Figure 5) data is coincidental because many regions should be on a later portion of the capture trajectory shown by Konikow and Leake (2014). Testing the model against observations would require more careful consideration of initial conditions and choice of simulated period. This may not be possible given the data available. However, presenting the simulation as an illustration of what would happen if pumping started in the year 2000 with no prior development is still a powerful demonstration of the capabilities of this model.

*We apologize for the misunderstanding that we seem to assume that not depletion occurred before 2000. This is not so. We have compared with GRACE only for the areas where we have that $q > q_{crit}$, while assuming that in that case $t > t_{crit}$ (exactly because we assume previous groundwater development). Thus, we compare depletion rates under this assumption with observed depletion rates from GRACE. We will make this more clear in the next version of the paper.*

It is not surprising that this approach reproduces similar patterns to other global models of groundwater depletion (lines 315-320). The assumptions and approaches are not that different in the models mentioned. A comparison of the results presented here to large-scale numerical models may provide a better test of model performance. Condon and Maxwell's (2019) model examining the impacts of groundwater pumping on streamflow over a large section of the USA at a 1 km resolution provides such an opportunity. There are assuredly some differences in computation times but the numerical approach will likely be

superior in resolving hydraulic gradients and could likely be done at a global scale in the near future.

*We believe it will be some time before the results of the model of Condon and Maxwell (2019) will be applied globally. Also, our approach is meant to make quick inferences and be used in e.g., regional hydroeconomic simulation and optimization approaches, requiring close to instantaneous results when applied. Nevertheless, our approach should be sufficiently trustworthy. Therefore, also following the suggestions of reviewer #3, we will extend the validation of our results. We intend to compare the aquifer and streamflow depletion (rates) with a) observations from and affected aquifer (e.g., Ogallala) and 2) with large-scale numerical models. We intend to compare to the results to a global groundwater model (de Graaf et al., 2019). We will contact Laura Condon to see if we can obtain the aquifer and streamflow reduction results (Figure 2A and 2B) of Condon and Maxwell (2019).*

Furthermore, the analytical approaches reviewed by Zipper et al. (2019) are not restricted to single wells, as suggested in line 116. Invoking superposition with some of those concepts may provide another path forward to study capture and depletion at large scales.

*This is a valid point, and we agree that it could be used by analyzing multiple wells by assuming superposition. We will acknowledge this in the next version of the paper. It remains however not possible to include the change in groundwater-surface water interaction from connected to disconnected in their approach.*

It is unclear that the approach used in the current study is "likely the simplest analytical form that can be devised to describe the effects of groundwater pumping at the larger scale" (lines 437-438). Objectively deciding the level of detail that effects of pumping need to be captured does not seem possible. Rather than making such claims, a more in-depth consideration of how the global approach presented here compares to numerical models or analytical techniques at local and regional scales might provide important context for this work. Such a comparison may help to guide future efforts in advancing large-scale groundwater modelling.

*Reviewer #1 also took issue with this this claim and we will delete this sentence from the next version of the paper. We hope that a more extensive validation in the following version of the paper will help to decide whether our lumped conceptual model is not too simple to provide reasonable estimates of aquifer and streamflow depletion at larger scales.*

This is a potentially important study in understanding large-scale groundwater depletion. While there are unresolved questions on the effectiveness of this approach is due to issues with initial conditions in the simulation, qualitatively it looks promising. The relationship between the results presented here and the threshold between sustainable and unsustainable development of groundwater is debatable. However, disconnection of water tables and streams is a clear indicator that groundwater pumping has resulted in an undesirable outcome and other thresholds may have already been passed.

*We thank the reviewer for this encouraging final paragraph of his review.*

**References**

Condon, L. E. and Maxwell, R. M.: Simulating the sensitivity of evapotranspiration and streamflow to large-scale groundwater depletion, Science advances, 5(6), eaav4574, 2019.

de Graaf, I.E.M., Gleeson, T., van Beek, L.P.H., Sutanudjaja, E.H. and Bierkens, M.F.P. (2019). Environmental flow limits to global groundwater pumping. *Nature* 574, 90-108.

Ferguson, G., Cuthbert, M. O., Befus, K., Gleeson, T., & McIntosh, J. C. (2020). Rethinking groundwater age. *Nature Geoscience*, 13(9), 592–594

---

## Author Comment (AC3) · 12 Feb 2021

**Reply to comments by anonymous reviewer #3**

*We thank reviewer 3 for his/her thorough review of our paper, which will help to improve the manuscript and the underlying study for the HESS readership. We will go over his/her comments point by point, with the comments in roman and our reply in italics. The specific actions we intend to perform in order to improve the paper are underlined.*

Bierkens et al. present a simplified analytical methodology for first order approximation of the impacts of groundwater pumping on streamflow and ultimately groundwater sustainability. In addition to summarizing the methodology they provide global mappings of streamflow depletions and sustainable pumping limits. While I appreciate that the authors were very clear throughout the manuscript that this methodology is intended to be approximate, I still have very significant concerns and I don't feel that the manuscript in its current form has demonstrated that this approach is adequate to support the types of groundwater sustainability findings that are presented in figures 7-9 for the following reasons:

1. The approach presented here relies on a myriad of simplifying assumptions. While the authors do try to be very transparent in these assumptions, this does not make them less concerning. Specifically, the steady state approach and the distributed well locations are big areas of concern in my opinion. For large scale aggregated analyses of declines this might be okay but for stream aquifer interactions well placement and timing is very important. The key advance of this paper is groundwater surface water interactions and therefore I think the bar is higher for some of these assumptions.

*We thank the reviewer for recognizing that we are clear about underlying assumptions. At the same time the reviewer is concerned whether the assumptions are such that no credible large-scale assessments can be made of the impact of groundwater pumping on streamflow. We start by saying that although well placement with respect to surface water is important at local scales, we argue that if many wells are considered, some are closer and other further away from the surface water bodies and that this may be accounted for in a lumped conceptual model aiming at describing the regional response (compared with lumped rainfall-runoff models for entire catchments) as developed in our paper. However, we understand this concern and we will add additional validation datasets (see hereafter), also related to the impacts on surface water, to assess the degree of applicability of our approach.*

2. The authors present a sensitivity analysis for their approach which is a helpful illustration of the relationship between variables. However, for me this really only demonstrates that the general interactions are in the correct direction, which follows directly from the equations they used. Much more concerning to me is the uncertainty of the inputs to these equations at the spatial scales presented here and whether reliable estimates for some of the parameters can be generated at all. For example, how accurately can bed slope and bed elevation be captured at this resolution globally? How sensitive are the final results to the uncertainty in these values?

*We still feel that the current local sensitivity analysis is insightful, as some relationships may follow from the equations in hindsight, but the behavior can nevertheless be rather complex, which is illustrated by the analysis. Regarding the uncertainties: the slope of the stream is not part of our model, but the stream-bottom elevation is. The uncertainty about stream bottom elevations is however not only a problem for our approach. It is also an issue for groundwater modelling at regional scales even. In most cases, even in regional modelling studies, but certainly for global modelling (Schulze et al., 2006; Sutanudjaja et al., 2018), stream dimensions are taken from geomorphological laws relating stream width and stream depth to bankfull discharge (Leopold and Maddock, 1953), which is the yearly maximum discharge with a return period of 2-3 years. In the parameterization for the global application, we use a stream dimension data-set from PCR-GLOBWB 2 (Sutanudjaja et al., 2018) that is based on this approach. To investigate the impact of uncertainty of bottom elevation d and the other uncertain parameter C, we will add a global sensitivity analysis elucidating the sensitivity of the results in Figures 4 and 6-8 to these parameters.*

3.  My biggest concern here is that the most important metrics that the authors are highlighting in their findings are not well validated. The authors present primarily comparisons to other global models which rely on similar assumptions and are working at similar spatial resolutions. It seems like it should be expected that the results here would be 'remarkably similar' (line 315-318). Before jumping to a global analysis I would like to see some rigorous evaluation of the model in some of the many heavily studied aquifers across the world comparing to regional models and observations. For example, observational groundwater ranges are reported for a few aquifers (Lines 379-382) but the authors only note that 'our estimates are in the lower end of those observed ranges' I think a much more quantitative comparison is need here.

*We agree that a further critical evaluation of the global results would be beneficial to scrutinize our global results. In the following version of the paper we intend to add a separate paragraph that compares the global results to: a) observed time series of groundwater levels and streamflow for part of an aquifer that is known for heavy groundwater exploitation (e.g., part of the Ogallala aquifer); b) to a regional high-resolution model study that produces both streamflow depletion estimates as well as groundwater taken out of storage as a result of pumping; c) to groundwater depletion rates and streamflow depletion rates as calculated by a global groundwater model (de Graaf et al., 2019).*

4.  Furthermore the validation that is provided here is really focused on groundwater depletions and I think the validation of the stream aquifer interactions or sustainable limits to groundwater withdrawals (the highlight of the paper) is lacking. If the main purpose of this work is to get to sustainability estimates and to connect to streamflow then these are the parts of the methodology which must be most thoroughly evaluated. I realize that this information is not available globally (hence the novelty of this work). However, I don't see any reason why these behaviors cannot be rigorously and quantitatively evaluated in some example locations for which data or models are available.

*Directly validating the limits, e.g,. $q_{crit}$, $t_{crit}$, $q_{eco}$ is not possible, regardless which approach is used to assess these. These limits will always be based on model estimates, regardless of the model used. Perhaps a local pumping experiment could provide local estimates, but otherwise they are non-observables that nevertheless provide guidelines to sustainable groundwater withdrawal when obtained with local models and threshold values to compare regional-scale pumping rates with in large-scale groundwater status assessment. However, we are able to provide sensitivity analyses (see answer to point 2 above) and additional evaluation of both groundwater and streamflow depletion or depletion rates (see point 3 above) to evaluate the underlying (lumped) conceptual model framework.*

5. Finally, the sustainability language in this paper is concerning to me. First of all because sustainability is a very subjective topic and it's not clear that the first order type approximations used here can really get at true sustainability. Second of all because I think these results can easily be misinterpreted based on how they are presented here. The authors do try to specify that this approach is only for first order approximations, but if that is the goal here then I think they should focus on using this methodology to provide ranges of potential groundwater depletions and stream interactions, and not be using this to present things link groundwater limits which can very easily be misinterpreted and miss-used.

*In accordance with the objections of the other reviewers about our limited notion of sustainability (mostly physically defined), we will refrain from using the term in our paper in relation to stream-groundwater disconnection. We believe that the limits presented in Figure 9 are still of use, because they are not to guide local withdrawal rates, but rather can be used in global studies on the state of groundwater use. We will provide this caveat in the next version of the paper.*

Overall, I do think this is a well written paper that is clearly presented and easy to follow. Unfortunately, I am not convinced about the validity of the approach, and as a result the findings that are presented. I think a much more rigorous evaluation of the methodology is needed including quantitative analysis of every metric that is going to be presented in the findings. I completely understand that this methodology is intended to be approximate and will not perform as well as regional integrated models or intensive observational studies. However, I think these should still be the bar for comparison so that users of this approach can fully understand its strengths and weaknesses of the simplified approach, and so that any metrics that are too uncertain are not included

*We thank the reviewer for the complements and acknowledge his/her concerns that echo what has been stated under points 1-5. We are certain that by providing additional sensitivity analyses, scrutinizing the global results with a more rigorous evaluation with data and the results of regional models and by removing the frame of groundwater sustainability, we will be able to address his/her concerns successfully.*

**References**

de Graaf, I.E.M., Gleeson, T., van Beek, L.P.H., Sutanudjaja, E.H. and Bierkens, M.F.P. (2019). Environmental flow limits to global groundwater pumping. *Nature* 574, 90-108.

Leopold, L. and Maddock, (1953). *The hydraulic geometry of stream channels and some physiographic implications*: Professional Paper 252, United States Geological Survey.

Schulze, K., Hunger, M and Doll, P. (2006).  Simulating river flow velocity on global scale. *Adv. Geosci*. 5, 133–136.

Sutanudjaja, E. H., van Beek, R., Wanders, N., Wada, Y., Bosmans et al. (2018). PCR-GLOBWB 2: a 5 arcmin global hydrological and water resources model. *Geosci. Model Dev*. 11, 2429–2453., 2018.

.

---

## Referee Report (RR1)

**Review of 'Large-scale sensitivities of groundwater and surface water to groundwater withdrawal' by Bierkens et al.**

*for Hydrology and Earth System Sciences*

**Summary**

Bierkens et al. present a novel analytical approach to assess the impact of groundwater withdrawals on the disconnection of streamflow and groundwater, that can be applied globally. The approach further facilitates the calculation of critical pumping rates that are needed to maintain the surface-groundwater connection. Results include an assessment of model parameters and qualitiative comparisons to other studies.

**Recommendation**

I am a new reviewer to this submission, which has already undergone a first round of major reviews. Thus, my comments focus on how well the authors have addressed the reviewer comments - but I am adding a few comments that I deem important and may require a few (minor) adjustments.

In general, I think that the authors addressed most (if not all) major comments from the reviewers. Major changes include, e.g., a reformulation of the manuscript focus (from sustainability to the disconnection of streamflow-groundwater interactions), additional validation and comparison exercise using other data sets (originating outside the group), additional supplementary materials were added, among other minor revisions to address the reviewer's concerns. There is just one overarching concern, raised by all reviewers, that could be addressed in a slightly better way. I would thus recommend that the authors revise their response to this concern in another minor revision.

**Issues with validation / intercomparisons**

All reviewers requested additional validation/comparisons to other studies. In response, the authors added (i) a comparison to other input data sets (i.e., critical transition times from Cuthbert et al), and (ii) qualitative intercomparisons to other global (de Graaf et al.), continental (Condon and Maxwell), and regional/local (Wen and Chen) studies. The latter includes a local validation with streamflow and groundwater head trends for the Republican River basin in the US.

While the authors went through a lot of effort employing these other data sets for the validation and intercomparison - which is appreciated -, I believe this is the only important part that may need clarification. While I understand that these

intercomparisons are difficult, especially since most of these variables cannot be measured, I think a bit more effort in explaining the differences and the expected shortcomings of the approach is needed.

This concerns, for example, the differences in critical transition times in Fig. 5 derived from the two data sets (Sutanudjaja et al.; and Cuthbert et al.). The differences here are between a factor 100 and a factor 1000. I am not an expert on this, but it is indeed "quite striking" (l. 399). The text does not really explain what the impact of these differences on the further results in the manuscript is (and if these are used at all). The explanation (l. 399-404) is not very convicing to me (but again, I am not an expert).

Further, the intercomparisons to other, e.g. numerical models such as ParFlow-CLM, remain rather qualitative. I would be okay with that if it was intuitive, but the text barely touches upon the reasons for differences. Instead, the comparisons are lacking behind. E.g., l. 546f: "Figure S12 shows again that the analytical approach yields larger depletion estimates than ParFlow, but the results are more similar than with de global model of De Graaf et al (2019)." ... but likely for different reasons, no? Also, for the comparison to de Graaf, the differences appear (maybe only) larger outside of the US.

Along these lines, personally, I think that a few more notes related to the shortcomings of the approach and its expected performance may be required at the end / in the summary. This is especially, because applications of this analytical approach with other data sets / models are encouraged. So I think the reader needs to know (i) why differences to the aforementioned studies appear, and (ii) where the model is expected to perform well and where not. I think this concern is in line with concerns from all reviewers (esp. #1 and #3), who criticize some of the assumptions - even though they are explicitly mentioned throughout the text.

**Some more minor notes**

- Please check the consistency of units throughout the text (e.g. use SI unit "d" instead of "day" and unify "yr", "y" and "year"
- l. 657-670: this should maybe be moved to the methods
- l. 104-105: check sentence; "these transitions do not occur" and "is that ... is that"
- l. 255: It should be "h_s(infinity)" I think
- l. 283: check sentence; "that are of interest to show" or remove "of"
- l. 360: and?
- l. 374: delete one "the"
- Fig. 6+7: could the authors add what is considered 'negligible' ?
- l. 545: "de" --> "the" (a little bit more Dunglish ;))
- Table S1: small v instead of large V for consistency with main manuscript?

**Concluding remark**

In general, I am very appreciative of the work and fully support the notion towards large scale hydrology. In particular, I welcome the broad applicability of the approach, e.g. at various scales and with other models/observations - and concur with the authors that this approach may be useful for (i) bridging the time gap until global numerical approaches are ready, but (ii) also for benchmarking, intercomparisons and uncertainty analyses.

---

## Author Response (AR2)

Dear Editor,

Thank you for the opportunity to revise our manuscript according to the remarks of two additional referees. They both raise valid points and we have changed the manuscript accordingly. This means that we:

- Reframed the manuscript in accordance with the suggestions of Referee 4, i.e., we now state explicitly in the abstract, the introduction and the discussion that the analytical framework should be paired with a more complex global hydrological model to be used as a fast and first-order screening tool.
- Added more discussion on the cause for differences between the critical transition times obtained from Sutanudjaja et al (2018) and Cuthbert et al (2019) and the validation results (Referee 5).
- Corrected the minor issues noted by Referee 5.

In the following we will respond point-by-point to the comments by the referees. The comments are denoted in Roman, our response in Italics and quotes of added text in the manuscript in Roman red. Line numbers refer to the newly revised manuscript.

We think that the two rounds of reviews have greatly improved the manuscript and hope that it is now ready for publication in HESS.

We await your decision with interest.

With kind regards,

Marc Bierkens (on behalf of my co-authors).

**Anonymous Referee 4**

*I am not a groundwater modeler. I have read the discussion of the paper (comments of the reviewers and the response of the authors) and the revised manuscript. This is an interesting discussion, and the authors have addressed many of the comments of the reviewers. Hopefully one or more of the previous reviewers will be satisfied with these revisions. I have taken a higher level view of the paper, and feel that a simple twist to the presentation of the paper will make the paper much more acceptable and appealing, and will could provide avenues for further extension of the work. This change will require some moderate revision of the paper.*

*My own view is that the simple model they present here is so simple and is such a caricature of groundwater theory that it cannot form the building block for a bottom-up distributed groundwater model. In fact, the application of the simple model in this paper draws its power from observations or detailed predictions by a more detailed and sophisticated model. In other words, the simple model can at best be a good screening tool to elaborate on controls on large-scale sensitivities of groundwater and surface water to groundwater withdrawal. In other words, the simple model is not a standalone model, but is a top-down model that is used in combination with a more detailed, physically based model such as PRC-GLOBWEB2 in this case.*

*When I first read the paper, I got the wrong impression that this is about development and validation of a simple groundwater model. The discussion and the response of the reviewers have clarified to me that this is wrong interpretation of the paper. However, a reframing of the paper along the lines I suggested above would give a more correct interpretation of the paper, and will bring out more the novelty and usefulness of the simple model as a screening tool to assess groundwater sustainability at regional scales. In this new reframing the pairing of the simple model and the more complex model (PRC-GLOBWEB) presents a analysis framework for assessment of groundwater sustainability, and there is every opportunity to refine and improve both kinds of models as more data and observations become available.*

*In any case, I hope the authors and the previous reviewers agree with or appreciate this interpretation and this can contribute to a further moderate revision of the paper, which will make it much more appealing to reviewers and readers alike.*

We thank anonymous referee 4 for the supportive review and valuable suggestions to stage the paper differently. We respectfully disagree with referee 4 that the analytical framework is a caricature of groundwater theory, as there have been previous publications showing that at larger scales surface-water groundwater interaction behaves as a (piecewise) linear reservoir (see e.g., Savenije (2018), Hydrol. Earth Syst. Sci., 22, 1911–1916) and a similar parameterization is used in MODFLOW and several global hydrological models. Nevertheless, its lumped nature makes it difficult to directly link the C-value to flow geometry and hydraulic properties of the aquifer (although our appendix shows an example on how it can be). That being said, we agree with the suggestions of the reviewer, and we have reframed the analytical framework and present is a screening tool to be used together

with a more complex model to quickly perform sensitivity studies focussed on regional-scale groundwater withdrawal impacts and sustainability. We have added the following lines:

Abstract lines 34,35: After a local sensitivity analysis, the framework is combined with parameters and inputs from a global hydrological model and subsequently used to provide global maps of critical withdrawal rates and timing, the areas where current withdrawal exceeds critical limits, and maps of groundwater depletion and streamflow depletion rates that result from groundwater withdrawal.

Abstract lines 39-41: Pairing of the analytical framework with more complex global hydrological models presents a screening tool for fast first-order assessments of regional-scale groundwater sustainability, and for supporting hydroeconomic models that require simple relationships between groundwater withdrawal rates and the evolution of pumping costs and environmental externalities.

Introduction lines 132-134: We envision that such an analytical framework, when parameterized with parameters and inputs from a more complex global-scale hydrological model, can be used as a screening tool for fast first-order assessments of regional-scale groundwater sustainability, and for supporting hydroeconomic models that require simple relationships between large-scale groundwater withdrawal rates and the evolution of pumping costs and environmental externalities.

Results lines 361-365: It should be noted that our results are obtained at only a fraction of the computational costs of global hydrological models: a few minutes at a single PC compared to 2 days on a 48-core machine with PCR-GLOBWB at 5 arc-minutes. Thus, the sensitivity to changing pumping rates or changes in recharge under climate change can be quickly evaluated.

Discussion and conclusions lines 609-615: We have introduced an analytical framework based on a lumped conceptual model that intents to describe to what extent groundwater withdrawal affects groundwater heads and streamflow under changing regimes of groundwater-surface water interaction. By feeding the framework with the parameters and inputs from a more complex hydrological model (i.e., PCR-GLOBWB), it can be used as a screening tool for regional-scale groundwater sustainability. i.e., by providing a rich tableau of hydrologically and ecologically relevant outputs at very limited computational costs.

We hope that with this addition, the value of our approach becomes more clear and attractive to the reader, as was intended by the reviewer.

**Anonymous Referee 5**

*Summary*
*Bierkens et al. present a novel analytical approach to assess the impact of groundwater withdrawals on the disconnection of streamflow and groundwater, that can be applied globally. The approach further facilitates the calculation of critical pumping rates that are needed to maintain the surface-groundwater connection. Results include an assessment of model parameters and qualitative comparisons to other studies.*

*Recommendation*
*I am a new reviewer to this submission, which has already undergone a first round of major reviews. Thus, my comments focus on how well the authors have addressed the reviewer comments - but I am adding a few comments that I deem important and may require a few (minor) adjustments.*
*In general, I think that the authors addressed most (if not all) major comments from the reviewers. Major changes include, e.g., a reformulation of the manuscript focus (from sustainability to the disconnection of streamflow-groundwater interactions), additional validation and comparison exercise using other data sets (originating outside the group), additional supplementary materials were added, among other minor revisions to address the reviewer's concerns. There is just one overarching concern, raised by all reviewers, that could be addressed in a slightly better way. I would thus recommend that the authors revise their response to this concern in another minor revision.*

*Issues with validation/Intercomparison*
*All reviewers requested additional validation/comparisons to other studies. In response, the authors added (i) a comparison to other input data sets (i.e., critical transition times from Cuthbert et al), and (ii) qualitative intercomparisons to other global (de Graaf et al.), continental (Condon and Maxwell), and regional/local (Wen and Chen) studies. The latter includes a local validation with streamflow and groundwater head trends for the Republican River basin in the US.*
*While the authors went through a lot of effort employing these other data sets for the validation and intercomparison - which is appreciated -, I believe this is the only important part that may need clarification. While I understand that these intercomparisons are difficult, especially since most of these variables cannot be measured, I think a bit more effort in explaining the differences and the expected shortcomings of the approach is needed.*

Thank you for the clear summary of our approach and the appreciation of the efforts we did to evaluate the approach during the last revision. We have added additional text trying to explain the differences between our approach and other models in the Results section and also additional notes about the applicability of the approach.

*This concerns, for example, the differences in critical transition times in Fig. 5 derived from the two data sets (Sutanudjaja et al.; and Cuthbert et al.). The differences here are between a factor 100 and a factor 1000. I am not an expert on this, but it is indeed "quite striking" (l. 399). The text does not really explain what the impact of these differences on the further*

*results in the manuscript is (and if these are used at all). The explanation (l. 399-404) is not very convincing to me (but again, I am not an expert).*

We have tried to explain in more depth what is the cause of the difference between the Sutanudjaja et al, (2018) and Cuthbert et al. (2019) results. We have also extracted a conclusion about the limitations of using the approach in estimating critical transition and e-folding times.

Results lines 374-382: These differences can even add up to 2-3 orders of magnitude, which is extremely large. The reason is that the characteristic response times based on Cuthbert et al. (2018) are much larger (also up to 2-3 orders of magnitude) than those based on PCR-GLOBWB. Since the e-folding time in the stable regime is close to proportional to the *C*-value (e.g., Figure 3g), this is also true for the critical transition time. The very large differences in response times between these two datasets reveals that our method is only as good as its inputs and that critical transition times and times to full capture calculated with our approach should be interpreted with care and as order of magnitude estimates at best.

*Further, the intercomparisons to other, e.g., numerical models such as ParFlow-CLM, remain rather qualitative. I would be okay with that if it was intuitive, but the text barely touches upon the reasons for differences. Instead, the comparisons are lacking behind. E.g., l. 546f: "Figure S12 shows again that the analytical approach yields larger depletion estimates than ParFlow, but the results are more similar than with de global model of De Graaf et al (2019)." ... but likely for different reasons, no? Also, for the comparison to de Graaf, the differences appear (maybe only) larger outside of the US.*

Although it remains speculation without deep insights into the models of Condon and Maxwell (2019) (which we do not have) to explain the differences, also more quantitatively. Nonetheless, we have added additional text providing a possible explanation on why the results of Condon and Maxwell are closer than those of the De Graaf et al. (2019). The most likely explanation is that, apart from neglecting the lateral flow between cells, which is taken into account by both Condon and Maxwell (2019) and De Graaf et al. (2019), our approach also neglects the falling dry of water courses, which is taken into account by Condon and Maxwell (2019) but not by the De Graaf et al. (2019). So, the first omission results in overestimation of the groundwater level decline and the second omission by an underestimation, which therefore partly offsets the overestimation in case of Condon and Maxwell (2019).

Results lines 511-518: It is speculative at best to explain why the results of Condon and Maxwell (2019) are more similar. One possible explanation may be that the overestimation of decline rates due to ignoring lateral flow between cells in our approach is partly offset by the neglect of headwater streams falling dry under continuous pumping. This effect is included in ParFlow-CLM, which results in larger head decline rates that are closer to ours. The global groundwater model of De Graaf et al (2019) does not include this effect as streams in this model remain water carrying, even if the groundwater level drops below the stream bottom elevation.

*Along these lines, personally, I think that a few more notes related to the shortcomings of the approach and its expected performance may be required at the end / in the summary. This is especially, because applications of this analytical approach with other data sets / models are encouraged. So I think the reader needs to know (i) why differences to the aforementioned studies appear, and (ii) where the model is expected to perform well and where not. I think this concern is in line with concerns from all reviewers (esp. #1 and #3), who criticize some of the assumptions - even though they are explicitly mentioned throughout the text.*

Comparison of our results to other global and regional results does not reveal geographic differences between our framework's accuracy. We can however say something about the minimum scale (resolution) the approach is still producing reasonable results and also about the type of variables that can be estimated at what accuracy. For an explanation of the cause of differences with other approaches we refer to the earlier comments.

Discussion lines 621-630: The estimated global groundwater and surface water depletion rates were compared with observations and model results at various scales (support and extent), with mixed but overall favourable results up to the sub-basin scale. Results show that the analytical framework provides similar results to that of global hydrological models, but tends to overestimate the groundwater depletion rates when compared to groundwater flow models that account for lateral flow between cells. Also, without calibration, the critical transient times, i.e., the time from commencement of pumping till the detachment of the water table from the stream, as well as the related time to full capture, are order-of-magnitude estimates at best. Finally, when using global datasets, the analytical framework is limited to the sub-basin scale and too coarse for local-scale estimates.

Some more minor notes
- *Please check the consistency of units throughout the text (e.g. use SI unit "d" instead of "day" and unify "yr", "y" and "year"*
Thanks for noticing this. We have corrected it.

- *l. 657-670: this should maybe be moved to the methods*
We have considered moving it to the Methods, but, since this is actually a further elaboration of the results and an example of application, we feel it better fits the results section. Therefore prefer to leave it where it is.

- *l. 104-105: check sentence; "these transitions do not occur" and "is that ... is that"*
Thank you for noticing. We have corrected this.

- *l. 255: It should be "h_s(infinity)" I think*
Line 255 refers to Table 1. It already shows h_s(infinity) in the table.

- *l. 283: check sentence; "that are of interest to show" or remove "of"*
Corrected

- *l. 360: and?*

We are sorry, but it is not clear what is meant by this note.

- *l. 374: delete one "between"*

Corrected

- *Fig. 6+7: could the authors add what is considered 'negligible' ?*

We have added "($< 10^{-4}$)" to quantify what we consider to be negligible.

- *l. 545: "de" --> "the" (a little bit more Dunglish ;))*

We corrected this piece of Dutch creeping in.

- *Table S1: small v instead of large V for consistency with main manuscript?*

The capital V stands for any of the named output variables in Table S1, so it is kept as is.

Concluding remark

*In general, I am very appreciative of the work and fully support the notion towards large scale hydrology. In particular, I welcome the broad applicability of the approach, e.g. at various scales and with other models/observations - and concur with the authors that this approach may be useful for (i) bridging the time gap until global numerical approaches are ready, but (ii) also for benchmarking, intercomparisons and uncertainty analyses.*

We thank the reviewer for the kind words and for the effort to make this a better paper.